# Underestimated Privacy Risks for Minority Populations in Large Language Model Unlearning

## Abstract

Large Language Models (LLMs) embed sensitive, human-generated data, prompting the need for unlearning methods. Although certified unlearning offers strong privacy guarantees, its restrictive assumptions make it unsuitable for LLMs, giving rise to various heuristic approaches typically assessed through empirical evaluations. These standard evaluations randomly select data for removal, apply unlearning techniques, and use membership inference attacks (MIAs) to compare unlearned models against models retrained without the removed data. However, to ensure robust privacy protections for every data point, it is essential to account for scenarios in which certain data subsets face elevated risks. Prior research suggests that outliers, particularly including data tied to minority groups, often exhibit higher memorization propensity which indicates they may be more difficult to unlearn. Building on these insights, we introduce a complementary, minority-aware evaluation framework to highlight blind spots in existing frameworks. We substantiate our findings with carefully designed experiments, using canaries with personally identifiable information (PII) to represent these minority subsets and demonstrate that they suffer at least 20% higher privacy leakage across various unlearning methods, MIAs, datasets, and LLM scales. Our proposed minority-aware evaluation framework marks an essential step toward more equitable and comprehensive assessments of LLM unlearning efficacy.

## 1 Introduction

Large Language Models (LLMs) are trained on vast and diverse datasets, often sourced from public content on the web, much of which is generated by humans (Touvron et al., 2023; Ouyang et al., 2022). This practice raises significant ethical concerns, particularly when the data includes sensitive information, leading to potential privacy violations. Individuals whose data has been used may seek to exercise their "right to be forgotten", a protection guaranteed by regulations such as the General Data Protection Regulation (GDPR) (Krzysztofek, 2018).

The ideal approach to fulfilling such a request is to retrain the LLM from scratch, excluding the data to be removed. However, this solution is prohibitively expensive and impractical for large-scale models. To address this, the concept of *machine unlearning* has emerged as a promising alternative. It seeks to efficiently modify the LLM so that it becomes statistically indistinguishable from a model retrained from scratch. In this way, no adversary could confidently determine whether a model has undergone an unlearning process or been retrained, ensuring compliance with the "right to be forgotten".

Unfortunately, it remains an open problem to enforce the formal unlearning guarantee for deep neural networks and LLMs without exact retraining. Despite recent progress in theoretical unlearning research (Guo et al., 2020; Sekhari et al., 2021; Neel et al., 2021; Ullah et al., 2021; Chien et al., 2023; Ullah & Arora, 2023; Chien et al., 2024a;b), their restrictive assumptions limit practical applicability to deep neural networks and LLMs. Concurrently, researchers have developed efficient unlearning heuristics and empirically evaluated their efficacy (Golatkar et al., 2020a;b; Graves et al., 2021; Liu et al., 2024a;c; Yao et al., 2024), often by comparing approximately unlearned models to those retrained from scratch (Pawelczyk et al., 2024a). Among the various evaluation methods, membership inference attacks (MIAs) (Shokri et al., 2017), originally developed to infer data usage during training, have been widely adopted for assessing unlearning performance (Shi et al., 2024b).

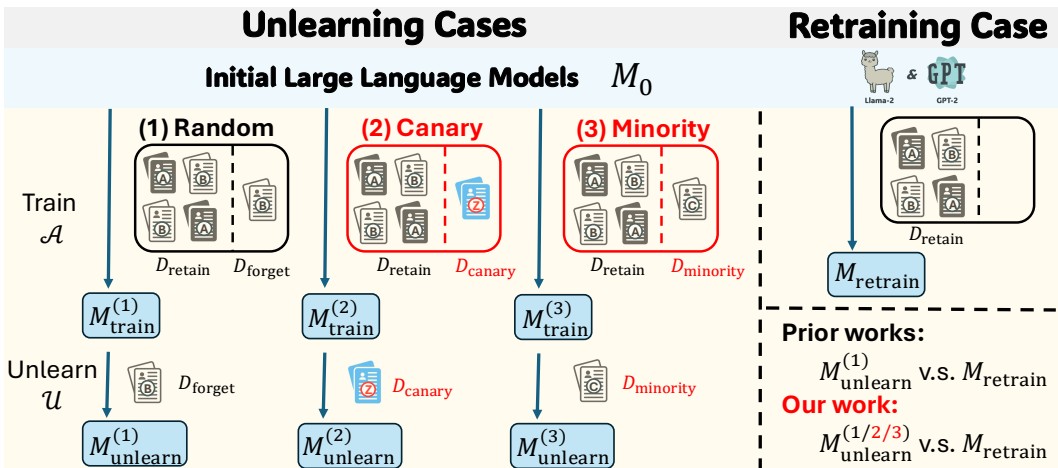

Figure 1: Illustration of our proposed LLM unlearning approaches (highlighted in red), when compared with the existing pipeline. Standard LLM unlearning evaluation typically involves randomly sampling data for removal from the training set (Case 1), which may underestimate privacy leakage for minority groups. In contrast, we design experiments to assess unlearning efficacy by removing canaries (deliberately inserted data points) related to minority groups (Case 2) and by directly removing data from minority groups (Case 3). Our approach provides a more comprehensive, minority-aware evaluation by considering the worst result across the three settings.

We identify a critical pitfall in the aforementioned LLM unlearning efficacy evaluation. While literature indicates that memorization levels in LLMs vary significantly across individual training samples (Feldman & Zhang, 2020; Carlini et al., 2022), current unlearning evaluation methods for them only capture "average-case" performance through random data removal from the training set. This approach inadequately addresses privacy risks for hard-to-unlearn data, failing to account for challenging scenarios necessitating rigorous privacy protection (Steinke & Ullman, 2020; Aerni et al., 2024). It neglects the principle that every individual's right to be forgotten should be upheld equally, thus ignoring data from minority groups, which are often treated as outliers and can be more resistant to unlearning due to the aforementioned stronger memorization effects (Carlini et al., 2022; Nasr et al., 2021; 2023). Consequently, standard unlearning evaluation significantly underestimates privacy risks for these groups, overlooking crucial social responsibilities in personal data protection.

**Contributions.** Motivated by the privacy auditing literature (Jagielski et al., 2020; Steinke et al., 2024), we conduct a synthetic experiment on unlearning injected canaries pertaining to minority groups. We choose Personally Identifiable Information (PII) as a representative minority identifier, while noting that our approach extends to broader cases. We show that minorities suffer from at least $20\%$ more privacy leakage in most cases across combinations of six unlearning approaches, three MIA variants, three datasets, and two LLMs of different scales. These results underscore the prevalence of the issue in practical settings, highlighting the need for a more effective LLM unlearning evaluation, particularly in regard to privacy risks for minority groups. Accordingly, we propose a *minority-aware LLM unlearning evaluation protocol* (Figure 1) as an initial step toward this goal. With this minority-aware protocol, we benchmark existing unlearning approaches and investigate the effects of forget set size as well as unlearning complexity. This study provides a more holistic understanding of different LLM unlearning approaches for practitioners. Notably, we observe that Langevin Unlearning—the only approach incorporating noise—achieves a favorable privacy-utility trade-off compared to noiseless methods such as SCRUB and Gradient Ascent (GA), suggesting a potential crucial role of noise incorporation in effective unlearning. In summary, these insights underline the critical role of our minority-aware evaluation framework in advancing equitable assessments of unlearning efficacy across different methods.

## 2 RELATED WORK

Privacy auditing is a fundamental yet challenging aspect of LLM unlearning due to the difficulty of distinguishing training samples effectively (Duan et al., 2024). Various privacy-related metrics, such

as exposure (Carlini et al., 2019), mean reciprocal rank (Wu et al., 2023), extraction likelihood (Jang et al., 2022), and truth ratio (Maini et al., 2024), have been proposed to probe privacy leakage. Among these, MIAs remain one of the most crucial tools for evaluating machine unlearning methods (Liu et al., 2024b). Standard MIAs typically involve training numerous shadow models independently to empirically approximate the distribution (Carlini et al., 2022). This approach has also been adopted for LLM unlearning, as seen in the NeurIPS 2023 Machine Unlearning Challenge[1] (which compares the point-wise output distributions of multiple unlearned and retrained models to perform MIAs) and Kurmanji et al. (2024); Pawelczyk et al. (2024b). Hayes et al. (2024) further highlights the limitations of average-case evaluations and introduces a specialized per-sample MIA method for unlearning evaluation. Their approach focuses on unlearning a randomly selected subset of training data by training a series of shadow models and performing per-sample MIA using a likelihood ratio test under Gaussian fitting (Carlini et al., 2022).

A major downside of MIA approaches involving shadow models is their computational expense, as they require training a large number of LLMs independently (Liu et al., 2024b). To address this downside, another line of research compares the outputs of models using different statistical metrics without requiring shadow models (Zhang et al., 2024; Liu et al., 2024a;c; Yao et al., 2024; Li et al., 2024), making these methods more computationally feasible (Maini et al., 2024). For instance, Shi et al. (2024b) measures privacy risk through the normalized AUC difference between unlearned and retrained models, using MIAs such as Min-K% (Shi et al., 2024a). However, these works typically select the forget set randomly from the training set, corresponding to an average-case evaluation. Our study highlights a critical limitation of this approach: the privacy risks of minority populations within the training set are severely underestimated because minority data are less likely to be selected in the unlearning evaluation pipeline. By focusing on minority-aware scenarios, our work provides a more comprehensive perspective on unlearning evaluation and privacy risks. Concurrently, Zhao et al. (2024) investigates how variations in the memorization of image representations impact unlearning performance. This aligns with our work, as we further demonstrate that, in text and natural language settings, minority data often suffer from a degradation in unlearning efficacy.

## 3 PRELIMINARIES

Machine unlearning (Cao & Yang, 2015; Bourtoule et al., 2021) has emerged as an important direction in trustworthy language models. It was initially motivated by privacy due to "the right to be forgotten" from GDPR and later extended to other legal and ethical concerns, including copyright (Yao et al., 2024), biased or outdated information mitigation (Liu et al., 2024b), hallucination removal (Yao et al., 2023), entity forgetting (Maini et al., 2024) and data poisoning removal (Pawelczyk et al., 2024a). In this work, we focus on the privacy aspect of the problem, albeit our methodology extends to other cases whenever the indistinguishability to the retrained model is an appropriate metric.

We briefly state the generic machine unlearning setting for privacy. Assume a training dataset $D_{\text{train}}$ and a holdout test set $D_{\text{test}}$ are given. Let $M_{\text{learn}} \leftarrow \mathcal{A}(M_0, D_{\text{train}})$ be the language model trained on $D_{\text{train}}$ starting from an initial model $M_0$ via the training algorithm $\mathcal{A}$, which may be either a pre-trained language model or random initialization. Once the model is trained, we receive data removal requests that partition the training set $D_{\text{train}} = D_{\text{forget}} \cup D_{\text{keep}}$ into a subset to be forgotten later $D_{\text{forget}}$ and a keep set $D_{\text{keep}}$. An unlearning algorithm $\mathcal{U}$ takes $M_{\text{learn}}, D_{\text{forget}}$ and $D_{\text{train}}$ as input to return an updated model $M_{\text{unlearn}} \leftarrow \mathcal{U}(M_{\text{learn}}, D_{\text{forget}}, D_{\text{train}})$. It is worth noting that $M_{\text{unlearn}}$ depends on the choice of $D_{\text{forget}}$. The gold standard to adhere to "the right to be forgotten" is retraining without $D_{\text{forget}}$, namely $M_{\text{retrain}} \leftarrow \mathcal{A}(M_0, D_{\text{train}} \setminus D_{\text{forget}})$. We say $\mathcal{U}$ achieves good unlearning efficacy if $M_{\text{unlearn}}$ and $M_{\text{retrain}}$ are indistinguishable in their behavior $m(M_{\text{unlearn}}, D) \approx m(M_{\text{retrain}}, D)$ on any corpus $D$, where $m$ is any evaluation metric. Since $M_{\text{unlearn}}$ and $M_{\text{retrain}}$ depend on the choice $D_{\text{forget}}$, such approximation should be taken over the worst case ideally.

### 3.1 EFFICIENT MIAS FOR LLM UNLEARNING

As previously discussed, the effectiveness of unlearning methods can be measured by the indistinguishability between the resulting unlearned models and an exactly retrained model. MIA is often leveraged to determine whether a specific sample is part of the training set and is widely applied to

---

[1]https://unlearning-challenge.github.io/assets/data/Machine_Unlearning_Metric.pdf

audit training data privacy leakage. Therefore, to evaluate the efficacy of an unlearning approach, we consider the **PrivLeak (PL)** metric (Shi et al., 2024b) defined as follows:

$$\textbf{PrivLeak (PL)} = \frac{\text{AUC}(M_{\text{unlearn}}; D_{\text{forget}}, D_{\text{test}}) - \text{AUC}(M_{\text{retrain}}; D_{\text{forget}}, D_{\text{test}})}{\text{AUC}(M_{\text{retrain}}; D_{\text{forget}}, D_{\text{test}})} \quad (1)$$

where AUC is the AUC-ROC score of an MIA (Ye et al., 2022) that tries to discriminate samples from $D_{\text{forget}}$ and $D_{\text{test}}$ based on the output statistics (e.g. loss) of a given model $M$. By normalizing the difference in AUC scores between $M_{\text{unlearn}}$ and $M_{\text{retrain}}$ using the AUC of $M_{\text{retrain}}$, the metric accounts for the inherent difficulty of distinguishing the forget and test sets. Note that for an effective unlearning method, the metric should be around zero since the behavior of $M_{\text{unlearn}}, M_{\text{retrain}}$ are indistinguishable. A larger magnitude of the PL metric implies a greater amount of privacy information that has been leaked under the tested MIA. A positive value indicates that the sample has not been fully forgotten, as the attacker has a higher AUC for $M_{\text{unlearn}}$ than $M_{\text{retrain}}$. Conversely, a negative metric value suggests over-forgetting, which still indicates that $M_{\text{unlearn}}$ differs from $M_{\text{retrain}}$ and thus cause privacy breaches. Finally, note that an effective unlearning solution should lead to a small PL metric for *any* choice of MIA. In this work, we consider three popular MIAs and report the corresponding PL metric.

- **lossMIA** (Yeom et al., 2018): Determines membership of a sample $x$ for a model $M$ based on its loss $\ell(M; x)$.
- **zlibMIA** (Carlini et al., 2021): Determines membership of a sample based on the sample loss normalized by its zlib compression size, $\ell(M; x)/\text{zlib}(x)$.
- **Min-K%** (Shi et al., 2023): Selects the lowest $K\%$ of token likelihoods and leverages the corresponding negative log-likelihood for membership inference.

## 4 THE UNDERESTIMATED PRIVACY RISK OF DATA MINORITIES

Recall that both the unlearned $M_{\text{unlearn}} \leftarrow \mathcal{U}(M_{\text{learn}}, D_{\text{forget}}, D_{\text{train}})$ and retrained $M_{\text{retrain}} \leftarrow \mathcal{A}(M_0, D_{\text{train}} \setminus D_{\text{forget}})$ language models depend on the choice of the forget set $D_{\text{forget}}$. Whenever we estimate the privacy leakage of an unlearning method $\mathcal{U}$ via some evaluation $m$, it is important to account for potential high-risk partitions of $D_{\text{forget}}$ to ensure a comprehensive assessment of privacy risk. Unfortunately, the current LLM unlearning evaluation pipeline overlooks this critical aspect, where the partition leading to $D_{\text{forget}}$ is chosen uniformly at random (Jang et al., 2023; Chen & Yang, 2023; Yao et al., 2024; Maini et al., 2024; Zhang et al., 2024; Shi et al., 2024b). The reported privacy risk therein hence corresponds to the "average case", which may significantly underestimate the privacy risk of highly privacy-sensitive points that request unlearning. It is known in the privacy literature that some rare training samples (minorities) may have an outsized effect on model memorization compared to common training samples (majorities) (Feldman & Zhang, 2020; Carlini et al., 2022). Intuitively, a similar phenomenon persists for unlearning.

Here we utilize the Enron dataset as a case study. This dataset comprises 535,703 authentic emails from 158 employees of the Enron Corporation. It is a standard benchmark dataset for studying PII leakage, where the phone number is one form of PII that has been extensively studied (Lukas et al., 2023). The phone numbers here follow the format of the U.S. phone numbers (e.g., 123-456-7890), with the first three digits serving as the area code, representing the location where the number holder applied for the number. Such information is considered sensitive as it leaks not only the phone number itself, but also the geographic information pertaining to the number holder.

Table 1: Top three most frequent and least frequent area codes within Enron dataset.

| Area code | Count |
|---|---|
| 713 (Houston) | 135,307 |
| 800 (Toll-free) | 11,902 |
| 212 (New York) | 10,739 |
| 484 (Allentown) | 1 |

Table 1 illustrates the least frequent and three most frequent area codes in the Enron dataset. The area code distribution is far from uniform. Consequently, if emails containing phone numbers are uniformly sampled for the forget set $D_{\text{forget}}$, minority data, such as emails with rare area codes like 484, are unlikely to be included due to their lower frequency. If unlearning minority data is inherently more challenging and results in greater privacy leakage, the existing evaluation pipeline may underestimate privacy risks for minorities.

### 4.1 VERIFY UNDERESTIMATED PRIVACY RISKS OF MINORITY VIA CANARY INJECTION

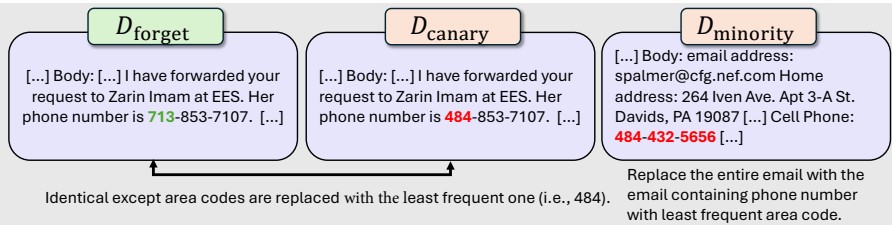

Figure 3: Illustration of the forget set $D_{\text{forget}}$, the construction of the canary set $D_{\text{canary}}$, and the minority set $D_{\text{minority}}$ for the Enron dataset. The minority set consists of emails with phone numbers containing the least frequent area codes. A histogram and distribution of area codes (e.g., 713 as the most frequent and 484 as the least frequent) are shown in Figure 2 and Table 1.

To rigorously show that *removing data from minority[2] populations indeed leads to higher unlearning privacy leakage*, we design experiments based on the idea of *canaries* in the privacy auditing literature (Jagielski et al., 2020; Steinke et al., 2024). For simplicity, we focus on the scenario where data removal requests pertain to PIIs, where each training sample $x \in D_{\text{train}}^{(1)}$ consists of PIIs such as phone numbers or organizations. We choose PIIs as a representative minority identifier, albeit a similar idea extends beyond PIIs. We consider the following cases, see Figure 1 for an illustration. 1) `Random`: we randomly partition $D_{\text{train}}^{(1)} = D_{\text{forget}} \cup D_{\text{keep}}$ as in the standard un-

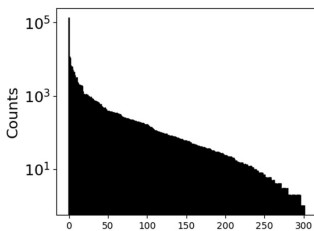

Figure 2: Area code histogram in Enron Dataset.

learning evaluation pipeline. This leads to $M_{\text{learn}}^{(1)} \leftarrow \mathcal{A}(M_0, D_{\text{train}}^{(1)})$.
2) `Canary`: For the same forget set $D_{\text{forget}} = \{x_i\}_{i=1}^n$, we construct a canary set $D_{\text{canary}} = \{x_i'\}_{i=1}^n$, where each $x_i'$ is *identical to $x_i$ except that only the PII is replaced by the least frequent one among $D_{\text{train}}^{(1)}$, isolating the impact of non-PII components*. Finally, we construct a synthetic training set $D_{\text{train}}^{(2)} = D_{\text{canary}} \cup D_{\text{keep}}$, which leads to $M_{\text{learn}}^{(2)} \leftarrow \mathcal{A}(M_0, D_{\text{train}}^{(2)})$. By executing the same unlearning evaluation process for both cases $M_{\text{learn}}^{(1)}$ and $M_{\text{learn}}^{(2)}$, we aim to show that the privacy risk for `Canary` is much higher than `Random`. By applying the same unlearning algorithm for removing $D_{\text{forget}}$ and $D_{\text{canary}}$, we obtain the unlearned model $M_{\text{unlearn}}^{(1)}$ and $M_{\text{unlearn}}^{(2)}$ respectively. The privacy leakage (PL) is then computed for these cases as described in Section 3.1. The calculation of PL for `Canary` entails replacing $D_{\text{forget}}$ with $D_{\text{canary}}$ in Eq. equation 1. Note that the retrained model $M_{\text{retrain}} \leftarrow \mathcal{A}(M_0, D_{\text{keep}})$ is identical for both scenarios.

An illustrative example of canary construction is provided in Figure 3. Note that for each email within $D_{\text{forget}}$ in `Random`, we construct the corresponding canary by only replacing its area code with the least frequent one (i.e., 484). This design is critical as we ensure the other part of the email is identical to the original email. Hence, if the privacy leakage of `Canary` is greater than `Random`, it must be due to the difference in the area code. We repeat the similar canary construction for the other PII such as email domain and year of legal judgment for different datasets.

## 4.2 QUANTIFY THE UNDERESTIMATED PRIVACY RISK OF UNLEARNING MINORITY

While our synthetic experiment on canary injection may be used to verify whether the unlearning privacy risk of minority populations is underestimated in the standard LLM unlearning evaluation pipeline, it cannot quantify the privacy risk for minorities in the real-world setting. We further design the third case aiming at quantifying the amount of underestimated privacy risk by directly choosing data to be removed containing the least frequent PII. 3) `Minority`: construct a set $D_{\text{minority}}$ that is of the same size as $D_{\text{forget}}$ in `Random`, which consists of samples with the least frequent PII within the dataset. By comparing the computed privacy risk of `Random` and `Minority`, we can quantify the amount of underestimated privacy risk for data removal from minority groups compared to the average case. If the resulting privacy risk is significantly higher than `Random`, any conclusion pertaining to unlearning efficacy drawn from `Random` can be misleading and the right to be forgotten of minorities is overlooked.

---

[2]*Minority* here refers to any subset of the data, defined by some shared value, that is under-represented in the training set. This term is generic and may apply to any type of attribute, demographic or otherwise.

# 5 UNLEARNING METHODS

We evaluate the following popular unlearning approaches in the literature. With a slight abuse of notation, we denote $M$ for both the model and its parameters for simplicity.

- **Random Labels (RL)** (Golatkar et al., 2020a; Yao et al., 2024): In the context of next-token prediction, this method involves randomly selecting tokens from the entire vocabulary during training on $D_{\text{forget}}$, aiming to disturb the model's learning from this dataset. The intuition behind this approach is that a model uninformed by $D_{\text{forget}}$ should behave as though it is randomly guessing the next token. However, as argued in Yao et al. (2024), this intuition may not always hold, depending on the specific scenario.

- **Exact Unlearning (EUk)** and **Catastrophic Forgetting (CFk)** (Goel et al., 2022): Exact unlearning can be done by retraining the entire model from scratch on $D_{\text{keep}}$, albeit it is prohibitively expansive in practice. Goel et al. (2022) proposes EUk method, which retrains only the last $k$ layers of the model while freezing the other layers. As a result, it is computationally cheaper than retraining the entire model. They also propose the CFk method, which continues training the last $k$ layers on the $D_{\text{keep}}$ without retraining from scratch, while freezing the other layers.

- **Gradient Ascent (GA)** (Golatkar et al., 2020a; Graves et al., 2021; Jang et al., 2023): Gradient ascent is arguably the most popular heuristic for machine unlearning. It seeks to remove the influence of the $D_{\text{forget}}$ from the trained model by reversing the gradient updates associated with $D_{\text{forget}}$. Notably, researchers have reported that gradient ascent can lead to significant model utility degradation in some cases (Ilharco et al., 2023; Pawelczyk et al., 2024a).

- **NegGrad+** (Kurmanji et al., 2024): NegGrad+ is a combination of gradient ascent on $D_{\text{forget}}$ and gradient descent on $D_{\text{keep}}$. It finetunes the current model by optimizing:

$$\beta \cdot \hat{\mathbb{E}}_{x \sim D_{\text{keep}}}[\ell(M; x)] - (1 - \beta)\hat{\mathbb{E}}_{x \sim D_{\text{forget}}}[\ell(M; x)]$$

where $\beta \in (0, 1)$ is a hyperparameter and $\hat{\mathbb{E}}$ is the empirical expectation. The intuition is to "review" the information from $D_{\text{keep}}$ in order to prevent the model degradation due to the gradient ascent.

- **SCRUB** (Kurmanji et al., 2024): SCalable Remembering and Unlearning unBound (SCRUB) is a state-of-the-art unlearning method that leverages a student-teacher framework. It updates the model by optimizing the objective:

$$\hat{\mathbb{E}}_{x \sim D_{\text{keep}}}[\text{KL}(M_{\text{learn}}(x)\|M(x)) + \ell(M; x)] - \hat{\mathbb{E}}_{x \sim D_{\text{forget}}}[\text{KL}(M_{\text{learn}}(x)\|M(x))]$$

where KL is the Kullback-Leibler divergence. SCRUB shares a similar intuition with NegGrad+, which can also be viewed as a combination of gradient ascent on $D_{\text{forget}}$ and descent on $D_{\text{keep}}$. Nevertheless, instead of directly employing the original loss $\ell$, SCRUB leverages the KL divergence to the original model $M_{\text{learn}}$. It provides a different regularization compared to NegGrad+.

- **Langevin Unlearning (Chien et al., 2024a;b)**: Langevin Unlearning leverages noisy gradient descent for machine unlearning. Specifically, during the training process, it replaces the common gradient descent with DP-SGD (Abadi et al., 2016). For unlearning process, it finetunes the model on $D_{\text{keep}}$ with DP-SGD as well. Chien et al. (2024a) establishes a smooth theoretical connection between differential privacy and unlearning and shows that Langevin Unlearning can provide a formal privacy guarantee for non-convex problems. Unfortunately, they mentioned that the resulting privacy bound is too loose to be applied in practice. We test Langevin Unlearning empirically in our experiments.

All the above unlearning methods fall under approximate unlearning (Thudi et al., 2022), valued for their practical efficiency. In contrast, exact unlearning methods, such as the sharding-based framework SISA (Bourtoule et al., 2021), demand significant computational and storage resources. Although SISA achieves exact unlearning by training multiple models independently on disjoint data partitions, this not only deviates from standard machine learning workflows but also introduces considerable memory overhead.

## 5.1 ENFORCING THE SAME COMPUTATION BUDGET FOR UNLEARNING METHODS

We categorize all methods as follows: those that only require the forget set (RL, GA), those that only require the keep set (EUk, CFk, Langevin), and those that require both the forget and keep

Table 2: The privacy leakage (PL) for each unlearning method against different attackers for GPT-2 / Llama-2 7B on the Enron-Phone dataset. The number in the parenthesis is the excess ratio of PL magnitude for cases `Canary` and `Minority` compared to `Random`, where a larger PL magnitude implies a more severe underestimation of privacy leakage in the standard evaluation (`Random`). Bold font indicates the case that the amount of underestimated privacy leakage is at least 20%.

| Method | PL (lossMIA) | | | PL (zlibMIA) | | | PL (Min-K%) | | |
|---|---|---|---|---|---|---|---|---|---|
| | Random | Canary | Minority | Random | Canary | Minority | Random | Canary | Minority |
| **Enron-Phone Dataset / GPT-2** | | | | | | | | | |
| No Unlearn | 0.190 | **0.283 (49%↑)** | **0.340 (79%↑)** | 0.052 | **0.076 (48%↑)** | **0.064 (24%↑)** | 0.300 | **0.447 (49%↑)** | **0.524 (75%↑)** |
| RL | 0.118 | **0.191 (61%↑)** | **0.210 (77%↑)** | 0.044 | **0.067 (52%↑)** | **0.060 (37%↑)** | 0.258 | **0.401 (55%↑)** | **0.447 (73%↑)** |
| EUk | 0.027 | **0.080 (198%↑)** | **0.124 (362%↑)** | 0.035 | **0.051 (47%↑)** | **0.052 (49%↑)** | 0.092 | **0.215 (134%↑)** | **0.223 (143%↑)** |
| CFk | 0.190 | **0.278 (46%↑)** | **0.337 (77%↑)** | 0.053 | **0.075 (41%↑)** | **0.064 (21%↑)** | 0.298 | **0.435 (46%↑)** | **0.514 (73%↑)** |
| GA | 0.089 | **0.140 (57%↑)** | **0.127 (42%↑)** | 0.024 | **0.042 (73%↑)** | 0.026 (7%↑) | 0.151 | **0.242 (60%↑)** | 0.171 (13%↑) |
| NegGrad+ | 0.183 | **0.271 (48%↑)** | **0.327 (79%↑)** | 0.052 | **0.073 (42%↑)** | 0.058 (13%↑) | 0.293 | **0.435 (48%↑)** | **0.511 (74%↑)** |
| SCRUB | 0.167 | **0.251 (50%↑)** | **0.321 (92%↑)** | 0.048 | **0.070 (44%↑)** | **0.062 (28%↑)** | 0.295 | **0.450 (52%↑)** | **0.527 (78%↑)** |
| Langevin | 0.093 | **0.144 (54%↑)** | **0.157 (69%↑)** | 0.024 | **0.037 (54%↑)** | 0.027 (12%↑) | 0.160 | **0.258 (61%↑)** | **0.264 (65%↑)** |
| **Enron-Phone Dataset / Llama-2 7B** | | | | | | | | | |
| No Unlearn | 0.060 | **0.242 (303%↑)** | **0.172 (187%↑)** | 0.034 | **0.098 (188%↑)** | **0.067 (97%↑)** | 0.076 | **0.115 (51%↑)** | **0.179 (136%↑)** |
| RL | -0.242 | -0.084 (65%↓) | -0.055 (77%↓) | -0.005 | **0.065 (1400%↑)** | **0.102 (2140%↑)** | -0.123 | -0.073 (41%↓) | 0.012 (90%↓) |
| EUk | 0.057 | **0.246 (332%↑)** | **0.185 (225%↑)** | 0.039 | **0.106 (172%↑)** | **0.082 (110%↑)** | 0.063 | **0.132 (110%↑)** | **0.189 (200%↑)** |
| CFk | 0.057 | **0.236 (314%↑)** | **0.168 (195%↑)** | 0.032 | **0.094 (194%↑)** | **0.063 (97%↑)** | 0.072 | **0.108 (50%↑)** | **0.171 (138%↑)** |
| GA | -0.562 | -0.430 (23%↓) | -0.464 (17%↓) | -0.014 | **0.038 (371%↑)** | **0.083 (593%↑)** | -0.625 | -0.459 (27%↓) | -0.517 (17%↓) |
| NegGrad+ | -0.074 | **-0.184 (149%↑)** | -0.040 (46%↓) | -0.021 | **-0.048 (129%↑)** | -0.002 (90%↓) | -0.069 | **-0.271 (293%↑)** | -0.057 (17%↓) |
| SCRUB | 0.059 | **0.162 (175%↑)** | **0.170 (188%↑)** | 0.034 | **0.063 (85%↑)** | **0.065 (91%↑)** | 0.074 | -0.031 (58%↓) | **0.177 (139%↑)** |
| Langevin | 0.033 | **0.180 (445%↑)** | **0.104 (215%↑)** | 0.016 | **0.068 (325%↑)** | **0.036 (125%↑)** | 0.033 | **0.055 (67%↑)** | **0.091 (176%↑)** |

sets (NegGrad+, SCRUB). Since machine unlearning is about the trade-off between privacy-utility-efficiency (Guo et al., 2020; Chien et al., 2024a; Liu et al., 2024c), we carefully ensure a similar computational complexity for all tested unlearning methods when demonstrating the privacy-utility trade-off. We define a **Complexity Unit** as the gradient computation budget of one training epoch on $|D_{\text{forget}}|$ samples and limit all unlearning methods to a maximum of 10 Complexity Units. Since $|D_{\text{forget}}| = U$ is roughly 1% of $|D_{\text{train}}|$ throughout our experiments, all unlearning methods are indeed much more efficient than retraining from scratch (Pawelczyk et al., 2024a).

For unlearning approaches that leverage $D_{\text{forget}}$ only, they can be tuned via unlearning process for at most 10 epochs. For those that leverage $D_{\text{keep}}$ only, we randomly subsample it to size $U$ for each epoch and unlearn for at most 10 epochs. For methods that leverage both $D_{\text{forget}}$ and $D_{\text{keep}}$ simultaneously, we limit their maximum unlearning epoch to 5. The situation is slightly more complicated for EUk and CFk approaches since only the last $k$ layers are trained to save computation. We randomly select $U/r$ samples from the keep set in each epoch, where $r$ is the ratio of trainable parameters in the last $k$ layers compared to the total number of parameters in the model. Our setup ensures that all tested unlearning approaches exhibit a similar unlearning computational complexity for a fair comparison. We optimize the unlearning epoch for each method under the 10 Complexity Unit constraint by the following criterion: if the perplexity of the unlearned model on $D_{\text{train}}$ increases by more than 1 point compared to that of the initial model (No Unlearn), we stop at the first epoch where this condition is met; otherwise, we use the checkpoint from the last epoch.

# 6 EXPERIMENTS

**Datasets.** Our LLM unlearning evaluation is conducted on two representative PII datasets: **Enron** (Klimt & Yang, 2004) and **ECHR** (Chalkidis et al., 2019). The Enron dataset contains corporate emails released by the Federal Energy Regulatory Commission, while the ECHR dataset comprises legal case information from the European Court of Human Rights. For our experiments, we focus on specific PIIs based on their distributions: **phone numbers** (Enron-Phone) and **email domains** (Enron-Email) in Enron, and the **year of judgment** (ECHR-Year) in ECHR. Data minorities are defined based on these PIIs. Detailed dataset statistics are provided in App. A.1. Our study centers on instance-level unlearning, treating each individual as a single record.

**General Settings.** We focus on the fine-tuning scenario, where the initial model $M_0$ is a pretrained LLM (GPT-2 (Radford et al., 2019) or Llama-2 7B (Touvron et al., 2023)). The fine-tuned model

$M_{\text{learn}}$ is obtained by training $M_0$ on a dataset $D_{\text{train}}$ for 5 epochs. In the GPT-2 experiments, both the training and test sets contain 10,000 samples, subsampled from the full dataset. For Llama-2, we employ efficient fine-tuning using LoRA (Hu et al., 2021); both the training and test sets consist of 50,000 samples, subsampled from the entire dataset. In all cases, the forget set size is set to 1% of the training set size. The models are optimized using the AdamW optimizer with a constant learning rate of $10^{-5}$, following the settings described in Shi et al. (2024b). During the unlearning process, all unlearning methods are constrained to the same computational budget—not exceeding 10 complexity units—as detailed in Section 5.1. We ensure that the unlearning complexity of each method is similar to allow for a fair comparison. Our ultimate goal is to achieve a superior privacy-utility-efficiency trade-off. We utilize MIA to estimate the empirical privacy risk measured by the PL metric as described in Section 3.1. For evaluating the utility of the LLMs, we report the perplexity following standard practices in the literature (Radford et al., 2019; Zhang et al., 2022), where a lower perplexity indicates that the model is more confident in its predictions. Additional details are provided in App. B.

## 6.1 STANDARD APPROACHES UNDERESTIMATE PRIVACY RISK FOR MINORITIES.

We report the results pertaining to the Enron-Phone, Enron-Email, and the ECHR-Year datasets. The experiment setting follows the explanation in Section 4 and further details are relegated to App. A.1. Table 2 shows that across all three attackers (lossMIA, zlibMIA, and Min-K%), all six unlearning methods and original model (no unlearning), the privacy leakage measure is significantly larger when unlearning canaries and minorities on Enron-Phone dataset for GPT-2 and Llama-2 7B respectively. Notably, in almost all cases the privacy leakage is underestimated for at least 20%. A similar phenomenon holds for the Enron-Email and ECHR-Year (Table 4, 5, 6, 7 in App. C.1) datasets. These results verify our claim that the current LLM unlearning evaluation indeed understated the privacy risk, especially for minorities. Our results call for a more careful empirical LLM unlearning evaluation, where considering canaries and minorities as we described can be an effective first step.

## 6.2 BENCHMARKING UNLEARNING APPROACHES UNDER MINORITY-AWARE EVALUATION.

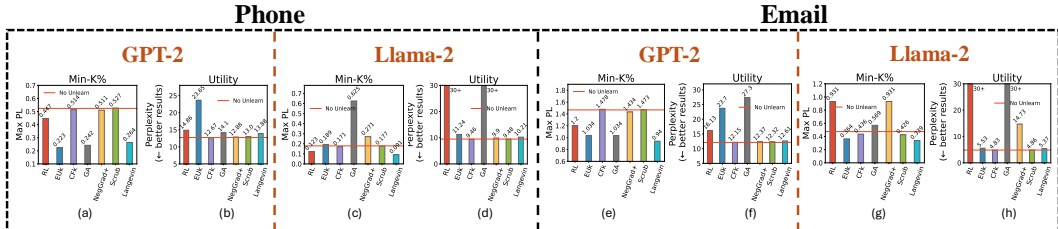

Figure 4: Benchmarking unlearning approaches via our minority-aware evaluation for GPT-2 and Llama-2 on Enron-Phone (Left) and Enron-Email (Right) datasets. (a),(c),(e),(g): Maximum privacy leakage (PL) over three cases (`Random`, `Canary`, and `Minority`) for Min-K% attack. (b),(d),(f),(h): Worst perplexity over the three cases of each method. More results on lossMIA and zlibMIA attackers are deferred to App. C.2.

Motivated by our observations, we propose the minority-aware LLM unlearning evaluation. Instead of reporting the privacy leakage (PL) score under the Random case, we propose to report the **magnitude of maximum PL score** of three settings (`Random`, `Canary`, and `Minority`). This provides a better privacy risk estimation while keeping the entire evaluation pipeline efficient. Besides, we report the corresponding worst-case perplexity as the utility measure for each unlearning approach. We benchmark the popular unlearning methods under our new evaluation pipeline, where the result is summarized in Figure 4 for GPT-2 and Llama-2 on the Enron-Phone and Enron-Email datasets. See App. C.2 for additional results.

We found that Langevin Unlearning offers the best balance between privacy and utility empirically. Note that while gradient ascent has on-par performance compared to Langevin Unlearning on the Enron-Phone dataset, it significantly degrades the model utility on the Enron-Email dataset. This echoes the finding of Ilharco et al. (2023); Pawelczyk et al. (2024a), albeit for different tasks. We found that gradient ascent is inherently unstable. In contrast, unlearning methods that leverage keep set $D_{\text{keep}}$ are much more stable, including Langevin Unlearning and SCRUB.

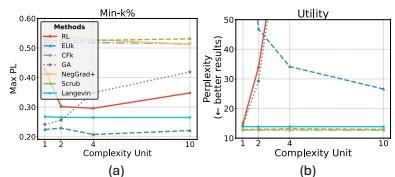 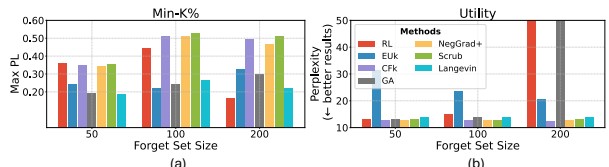

| The effect of unlearning epochs for each unlearning approach.(a): Maximum PL with the attacker being Min-K%. (b): Model perplexity. | The effect of forget set size for each unlearning approach.(a): Maximum PL with the attacker being Min-K%. (Results on loss-MIA, zlibMIA are deferred to App. C.5). (b): Model perplexity. |
|---|---|

Figure 5: Ablation studies on unlearning iterations and forget set size.

We also present results using utility metrics, including BERTScore (Zhang et al., 2019) and ROUGE (Lin, 2004), which capture semantic meaning. These results, provided in App. C.3, exhibit a consistent trend.

### 6.3 ABLATION STUDIES.

We present ablation studies on the Enron-Phone dataset using the GPT-2 model, unless otherwise specified, and further results are deferred to App. C.4, C.5 and C.6.

**Privacy-Utility Trade-off.** We analyze the privacy-utility trade-off curves for stable methods like Langevin Unlearning and SCRUB, along with the widely used GA. For Langevin Unlearning, we adjust the noise scale during training and unlearning. In SCRUB, we vary the weights that balance the loss and KL regularizer terms in its objective function (Sec. 5). For GA, we explore different learning rates ranging from $1e-7$ to $1e-3$. As shown in Fig.6, these curves are evaluated on the Enron-Phone and Enron-Email datasets. The results indicate that Langevin Unlearning (Green Line) outperforms SCRUB (Purple Line) with a superior privacy-utility trade-off. Notably, while GA performs reasonably well on the Enron-Phone dataset, its trade-off on Enron-Email is significantly weaker, revealing its instability. Detailed hyperparameter tuning for these methods is provided in App.C.6.

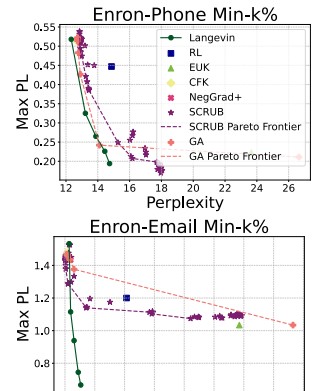

**Unlearning Iteration.** We investigate the effect of unlearning epochs on privacy (Max PL) and utility (perplexity) for each unlearning approach in Fig. 5 (Left). We observe that RL and GA are unstable in PL score. Furthermore, these two methods can lead to significant model utility degradation in terms of perplexity, where even unlearning for 2 epochs can already result in a model breakdown. This observation again demonstrates that gradient ascent, albeit being simple and popular, is not a reliable LLM unlearning solution. We should focus on stable unlearning solutions such as SCRUB and Langevin Unlearning.

Figure 6: Privacy-utility Trade-off Curves for GPT-2.

**Size of Forget Set.** In Fig. 5 (Right), we report the effect of different forget set sizes on the privacy (Max PL) and utility (Perplexity) trade-offs for each unlearning method. We find that both the GA and RL methods are highly sensitive to the forget set size, leading to significant model utility degradation and poor reliability in practice. In contrast, methods like Langevin Unlearning demonstrate good performance in terms of stability.

## 7 CONCLUSIONS

We identify a critical limitation in the typical evaluation pipeline for LLM unlearning efficacy: *privacy risks to minority groups in the training data are often underestimated*. Through carefully designed experiments using unlearning canaries tied to minority groups, inspired by privacy auditing research, we demonstrate that minority groups face at least 20% greater privacy leakage on average. Using personally identifiable information (PII) as a proxy for minority identifiers, we emphasize the need for more rigorous evaluations to ensure the right to be forgotten applies universally. Benchmarking existing unlearning methods with our minority-aware evaluation reveals that popular heuristics like gradient ascent are unstable and can degrade model utility. In contrast, methods such as Langevin Unlearning achieve a more favorable privacy-utility trade-off.

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

# Appendix

CONTENTS

# A  DATASET

## A.1  DATASET DETAILS

In this paper, we examine two representative PII datasets: Enron and ECHR, described as follows:

- **Enron (Klimt & Yang, 2004).** The Enron dataset consists of 536,000 authentic emails from 158 employees of the Enron Corporation, made publicly available by the Federal Energy Regulatory Commission following an investigation. Each email typically includes the sending timestamp, sender and recipient information, a greeting, the main content, and a footer containing the sender's personal details.

- **ECHR (Chalkidis et al., 2019).** The ECHR dataset comprises case records from the European Court of Human Rights. Each record contains a series of factual lists that detail the specifics of a case. In our experiments, we further decompose these cases into individual facts, with each fact forming a distinct sample, averaging around 80 tokens in length. In total, the dataset includes around 118,000 samples.

## A.2  PII SELECTION

As outlined in Sec. 4, we selected U.S. phone numbers from the Enron dataset based on a criterion aimed at analyzing privacy risks in minority groups. To ensure the PII distribution was imbalanced, reflecting both minority and majority groups, we additionally selected two standard PII types (Lukas et al., 2023): email addresses (Enron) and years (ECHR). The distributions of these PII counts are depicted in Fig. 7.

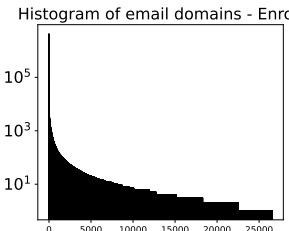
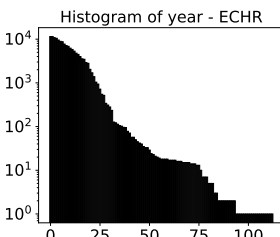

Figure 7: Histogram of email addresses (Enron-Email) and years (ECHR-Year).

## A.3  PREPROCESSING AND DATASET SPLIT

**Dataset Preprocess.** In our experiments, since the average token length of Enron samples is approximately 770, we controlled the token length of each fact to ensure the model could effectively memorize the samples. We randomly selected three coherent sentences from each sample, and if the sample contained specific PIIs of interest, we prioritized selecting sentences around them. We will keep the original samples for the ECHR dataset.

**Dataset Construction.** We begin by searching the dataset for occurrences of specific PIIs and analyzing their distribution. To form the minority set used in our `Minority` setting, we select 100 samples containing the least frequent PIIs; this set serves as our forget set. In the `Random` setting, we construct the forget set by randomly selecting 100 samples containing PIIs. To create the canary set, we replace the PIIs in the forget set (`Random` setting) with the least frequent PII found in the dataset. From the remaining data, we randomly select samples to create the training and test sets. For experiments with GPT-2 (117M), we uniformly at random selected 10,000 samples each for both the training and test sets. For Llama-2 7B, we uniformly at random selected 50,000 samples for both the training and test sets.

# B EXPERIMENTAL DETAILS

## B.1 COMPUTE CONFIGURATIONS

All experiments were conducted using 8 NVIDIA A100 GPUs (80GB) and 14 NVIDIA RTX 6000 Ada GPUs (48GB).

## B.2 UNLEARNING EXPERIMENT SETUP

**Unlearning Algorithms.** For all unlearning methods, we use a constant learning rate of $10^{-5}$ and a batch size of 32, consistent with the fine-tuning stage. Note that some unlearning algorithms require additional hyperparameters. We follow the common designs from previous literature (Pawelczyk et al., 2024a) and detail the hyperparameter selection as follows:

- **EUK and CFK.** In our experiments, we set the number of retrained layers to $k = 3$ for both GPT-2 and Llama-2 (LoRA) models. For GPT-2, the unfrozen trainable parameters account for approximately 16% of the total parameters, while for Llama-2 7B, the unfrozen parameters account for around 10%.

- **NegGrad+.** As noted in the main text, the hyperparameter $\beta$ balances samples between $D_{\text{forget}}$ and $D_{\text{keep}}$. In these experiments, we set $\beta = 0.999$.

- **SCRUB.** In the SCRUB method, three hyperparameters are used to balance the loss function on the keep set and the KL regularizers on both the keep and forget sets. According to the definition of the objective function in Section 5, three terms are weighted sequentially by setting: $\alpha = 0.5$, $\beta = 1$, and $\gamma = 0.01$.

- **Langevin Unlearning.** The Langevin Unlearning method leverages noisy gradient descent to unlearn samples from the forget set. In our experiments, we set the Gaussian noise scale to $\sigma = 5e - 4$ (for GPT-2) and $\sigma = 5e - 3$ (for Llama-2), and the clipping norm to 1.

**Unlearning Epoch Selection.** As outlined in Section 5.1, all unlearning methods are constrained to a maximum of 10 complexity units, and the optimal epoch for each method is selected based on whether the perplexity of the unlearned model on $D_{\text{train}}$ increases by more than 1 point. Under our computational budget, methods that only require the forget set (RL, GA) are run for 10 epochs, while methods requiring both the keep and forget sets (NegGrad+, SCRUB) are limited to 5 epochs, due to the equal-sized cycling between the two sets. For methods that only require the keep set (EUK, CFK, Langevin), we use 10 epochs, with varying sample sizes for EUK and CFK, as some model parameters remain frozen. The selected epoch for each method in each experiment is detailed in Table 3.

Table 3: Epochs comparison between unlearning methods on GPT2 and LLaMA2 models.

| Unlearning Methods | GPT-2 | | Llama-2 7B | |
|---|---|---|---|---|
| | **Enron** | **ECHR** | **Enron** | **ECHR** |
| **RL** | Epoch 1 | Epoch 1 | Epoch 1 | Epoch 1 |
| **EUk** | Epoch 10 | Epoch 10 | Epoch 10 | Epoch 10 |
| **CFk** | Epoch 10 | Epoch 10 | Epoch 10 | Epoch 10 |
| **GA** | Epoch 1 | Epoch 1 | Epoch 1 | Epoch 1 |
| **NegGrad+** | Epoch 5 | Epoch 5 | Epoch 1 | Epoch 5 |
| **SCRUB** | Epoch 5 | Epoch 5 | Epoch 5 | Epoch 5 |
| **Langevin** | Epoch 10 | Epoch 10 | Epoch 10 | Epoch 10 |

**Attack Method Hyperparameters.** We employed three attack methods in our evaluation pipeline. For lossMIA and zlibMIA, there are no hyperparameters to tune. The Min-$K\%$ method is based on the observation that non-member examples tend to have more tokens with lower likelihoods compared to member examples. In this method, the hyperparameter $K$ controls the selection of the bottom $K\%$ of tokens in each sample based on their likelihoods. Following previous recommendations in Duan et al. (2024); Shi et al. (2024a), we set $K = 20$ in our experiments.

Table 4: The privacy leakage (PL) for each unlearning method against different attackers for GPT-2 on the Enron-Email dataset.

| Method | PL (lossMIA) | | | PL (zlibMIA) | | | PL (Min-K%) | | |
|---|---|---|---|---|---|---|---|---|---|
| | Random | Canary | Minority | Random | Canary | Minority | Random | Canary | Minority |
| No Unlearn | 0.303 | **0.535 (77%↑)** | **1.145 (278%↑)** | 0.200 | **0.309 (54%↑)** | **0.262 (31%↑)** | 0.529 | **0.934 (76%↑)** | **1.468 (178%↑)** |
| RL | 0.033 | **0.153 (366%↑)** | **0.448 (1265%↑)** | 0.062 | **0.142 (127%↑)** | **0.121 (94%↑)** | 0.431 | **0.772 (79%↑)** | **1.200 (179%↑)** |
| EUk | 0.232 | **0.440 (89%↑)** | **0.582 (150%↑)** | 0.135 | **0.229 (70%↑)** | **0.152 (13%↑)** | 0.501 | **0.886 (77%↑)** | **1.034 (106%↑)** |
| CFk | 0.296 | **0.515 (74%↑)** | **1.139 (285%↑)** | 0.197 | **0.295 (49%↑)** | **0.260 (32%↑)** | 0.526 | **0.905 (72%↑)** | **1.478 (181%↑)** |
| GA | -0.279 | -0.173 (38%↓) | **0.739 (165%↑)** | -0.119 | -0.037 (69%↓) | **0.168 (41%↑)** | -0.390 | -0.304 (22%↓) | **1.034 (165%↑)** |
| NegGrad+ | 0.265 | **0.471 (77%↑)** | **1.103 (316%↑)** | 0.179 | **0.269 (50%↑)** | **0.251 (40%↑)** | 0.496 | **0.864 (74%↑)** | **1.434 (189%↑)** |
| SCRUB | 0.286 | **0.499 (74%↑)** | **1.097 (283%↑)** | 0.190 | **0.289 (52%↑)** | **0.253 (34%↑)** | 0.519 | **0.902 (74%↑)** | **1.473 (184%↑)** |
| Langevin | 0.154 | **0.319 (107%↑)** | **0.606 (293%↑)** | 0.086 | **0.178 (107%↑)** | **0.124 (44%↑)** | 0.336 | **0.645 (92%↑)** | **0.940 (180%↑)** |

Table 5: The privacy leakage (PL) for each unlearning method against different attackers for GPT-2 on ECHR-year datasets.

| Method | PL (lossMIA) | | | PL (zlibMIA) | | | PL (Min-K%) | | |
|---|---|---|---|---|---|---|---|---|---|
| | Random | Canary | Minority | Random | Canary | Minority | Random | Canary | Minority |
| No Unlearn | 0.198 | **0.247 (25%↑)** | **0.263 (33%↑)** | 0.086 | **0.103 (20%↑)** | **0.122 (42%↑)** | 0.213 | **0.276 (30%↑)** | **0.299 (40%↑)** |
| RL | 0.161 | **0.213 (32%↑)** | **0.234 (45%↑)** | 0.067 | **0.088 (31%↑)** | **0.086 (28%↑)** | 0.190 | **0.259 (36%↑)** | **0.257 (35%↑)** |
| EUk | 0.125 | **0.176 (41%↑)** | 0.138 (10%↑) | 0.067 | **0.088 (31%↑)** | 0.070 (4%↑) | 0.114 | **0.187 (64%↑)** | 0.135 (18%↑) |
| CFk | 0.188 | **0.234 (24%↑)** | **0.260 (38%↑)** | 0.084 | 0.095 (13%↑) | **0.120 (43%↑)** | 0.209 | **0.264 (26%↑)** | **0.295 (41%↑)** |
| GA | 0.067 | 0.027 (60%↓) | **0.105 (57%↑)** | 0.024 | 0.019 (21%↓) | **0.038 (58%↑)** | 0.090 | -0.019 (79%↓) | **0.143 (59%↑)** |
| NegGrad+ | 0.183 | **0.221 (21%↑)** | **0.247 (35%↑)** | 0.071 | **0.088 (24%↑)** | **0.112 (58%↑)** | 0.191 | **0.237 (24%↑)** | **0.274 (43%↑)** |
| SCRUB | 0.179 | **0.223 (25%↑)** | **0.253 (41%↑)** | 0.080 | **0.099 (24%↑)** | **0.116 (45%↑)** | 0.197 | **0.253 (38%↑)** | **0.289 (47%↑)** |

**Random Seed Selection.** In all our experiments, we followed the common practice and fixed our random seed to be 42.

# C    ADDITIONAL EXPERIMENTAL RESULTS

In this section, we present supplementary experimental results to further substantiate our claims in the main text.

## C.1    EXPERIMENTS ON ENRON-EMAIL AND ECHR-YEAR DATASETS

In Tables 4, 7, 5 and 7, we report the PL scores for all three attackers across the three scenarios on the Enron-email, ECHR-year datasets for GPT-2 and Llama-2, respectively. The results support our claim that the current LLM unlearning evaluation (`Random` setting) significantly underestimates privacy risk.

Table 6: The privacy leakage (PL) for each unlearning method against different attackers for Llama-2 7B on the Enron-Email dataset.

| Method | PL (lossMIA) | | | PL (zlibMIA) | | | PL (Min-K%) | | |
|---|---|---|---|---|---|---|---|---|---|
| | Random | Canary | Minority | Random | Canary | Minority | Random | Canary | Minority |
| No Unlearn | 0.050 | **0.282 (464%↑)** | **0.174 (248%↑)** | 0.046 | **0.236 (413%↑)** | **0.095 (106%↑)** | 0.064 | **0.474 (640%↑)** | **0.224 (250%↑)** |
| RL | -0.609 | -0.567 (7%↓) | -0.821 (12%↓) | -0.832 | -0.874 (5%↑) | -0.478 (43%↓) | -0.931 | -0.931 (0%) | -0.849 (9%↓) |
| EUk | 0.037 | **0.241 (551%↑)** | **0.169 (356%↑)** | 0.015 | **0.186 (1140%↑)** | **0.102 (580%↑)** | 0.040 | **0.364 (810%↑)** | **0.206 (415%↑)** |
| CFk | 0.049 | **0.264 (438%↑)** | **0.169 (245%↑)** | 0.046 | **0.220 (378%↑)** | **0.090 (96%↑)** | 0.062 | **0.436 (603%↑)** | **0.218 (251%↑)** |
| GA | -0.512 | **-0.692 (35%↑)** | 0.059 (89%↓) | -0.435 | -0.232 (47%↓) | 0.184 (58%↓) | -0.569 | -0.479 (16%↓) | 0.294 (48%↓) |
| NegGrad+ | -0.931 | -0.929 (2%↓) | -0.821 (12%↓) | -0.832 | -0.874 (5%↑) | -0.478 (43%↓) | -0.931 | -0.931 (0%) | -0.849 (9%↓) |
| SCRUB | 0.040 | **0.257 (543%↑)** | **0.174 (335%↑)** | 0.034 | **0.209 (515%↑)** | **0.095 (179%↑)** | 0.056 | **0.426 (661%↑)** | **0.224 (300%↑)** |
| Langevin | 0.022 | **0.191 (768%↑)** | **0.048 (118%↑)** | 0.020 | **0.141 (605%↑)** | 0.021 (5%↑) | 0.035 | **0.339 (868%↑)** | **0.079 (126%↑)** |

## C.2    MORE RESULTS ON MINORITY-AWARE EVALUATION

In this section, we present further benchmarking results for unlearning approaches under minority-aware LLM evaluation. Following the same setup as Section 6.2, Fig. 8 reports the maximum PL

Table 7: The privacy leakage (PL) for each unlearning method against different attackers for Llama-2 7B on the ECHR-year dataset.

| Method | PL (lossMIA) | | | PL (zlibMIA) | | | PL (Min-K%) | | |
|---|---|---|---|---|---|---|---|---|---|
| | Random | Canary | Minority | Random | Canary | Minority | Random | Canary | Minority |
| No Unlearn | 0.056 | **0.094 (68%↑)** | **0.076 (35%↑)** | 0.030 | **0.048 (60%↑)** | **0.096 (220%↑)** | 0.067 | **0.114 (70%↑)** | **0.138 (106%↑)** |
| RL | -0.069 | 0.044 (36%↓) | **-0.532 (671%↑)** | -0.024 | **0.029 (21%↑)** | **-0.192 (700%↑)** | -0.034 | **0.070 (106%↑)** | **-0.458 (1247%↑)** |
| EUk | 0.059 | **0.084 (42%↑)** | **0.079 (34%↑)** | 0.030 | **0.044 (47%↑)** | **0.079 (163%↑)** | 0.065 | **0.110 (69%↑)** | **0.153 (135%↑)** |
| CFk | 0.056 | **0.088 (57%↑)** | **0.073 (30%↑)** | 0.028 | **0.044 (57%↑)** | **0.088 (214%↑)** | 0.063 | **0.106 (68%↑)** | **0.131 (108%↑)** |
| GA | -0.046 | **-0.376 (717%↑)** | **-0.624 (1257%↑)** | -0.016 | **-0.120 (650%↑)** | **-0.267 (1569%↑)** | -0.063 | **-0.404 (541%↑)** | **-0.574 (811%↑)** |
| NegGrad+ | 0.024 | **-0.272 (1033%↑)** | **-0.624 (2500%↑)** | 0.012 | **-0.099 (725%↑)** | **-0.235 (1858%↑)** | 0.026 | **-0.404 (1454%↑)** | **-0.663 (2449%↑)** |
| SCRUB | 0.056 | **0.094 (68%↑)** | **0.073 (30%↑)** | 0.030 | **0.048 (60%↑)** | **0.090 (200%↑)** | 0.067 | **0.112 (67%↑)** | **0.139 (107%↑)** |
| Langevin | 0.026 | **0.052 (100%↑)** | **0.041 (58%↑)** | 0.010 | **0.025 (150%↑)** | **0.046 (360%↑)** | 0.028 | **0.062 (121%↑)** | **0.078 (179%↑)** |

score under lossMIA and zlibMIA attackers on Enron-Phone and Enron-Email and Fig. 9 reports the maximum PL score and worst-case perplexity for various unlearning methods on ECHR-Year (GPT-2) dataset.

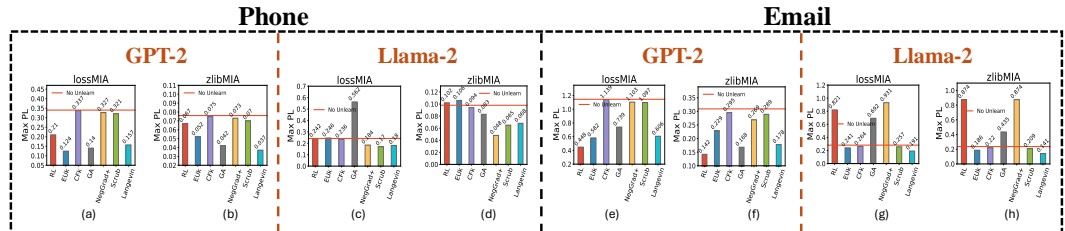

Figure 8: Benchmarking unlearning approaches via our minority-aware evaluation for GPT-2 and Llama-2 on Enron-Phone and Enron-Email dataset. (a),(c),(e),(g): Maximum privacy leakage (PL) over three cases (Random, Canary, and Minority) for lossMIA attack. (b),(d),(f),(h): Maximum privacy leakage (PL) over three cases (Random, Canary, and Minority) for zlibMIA attack.

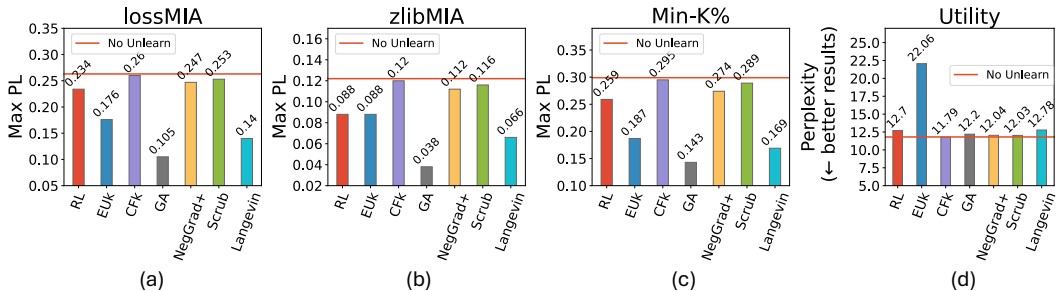

Figure 9: Benchmarking unlearning approaches via our minority-aware evaluation for GPT-2 on ECHR-year dataset. (a)-(c): Maximum privacy leakage (PL) over three cases (Random, Canary, and Minority) for lossMIA, zlibMIA, and Min-K% attacks respectively. (d): Worst perplexity over the three cases of each method.

We observe that both GA and Langevin Unlearning methods maintain a favorable balance between privacy and utility. However, GA can be sensitive to the forget set size and the number of unlearning iterations (Section 6.3). In practice, the GA method should be applied with caution, whereas more stable approaches like Langevin Unlearning offer a better trade-off in terms of privacy, utility, and stability.

## C.3 RESULTS ON OTHER UTILITY METRICS

In this section, we report the worst-case utility performance across three settings (Random, Canary, and Minority) using utility metrics BERTScore (Zhang et al., 2019) and ROUGE score (Lin,

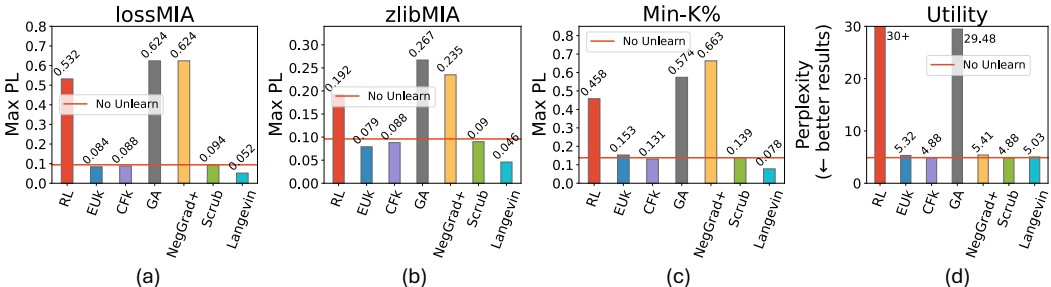

Figure 10: Benchmarking unlearning approaches via our minority-aware evaluation for Llama-2 on ECHR Year dataset. (a)-(c): Maximum privacy leakage (PL) over three cases (Random, Canary, and Minority) for lossMIA, zlibMIA, and Min-K% attacks respectively. (d): Worst perplexity over the three cases of each method.

2004), which capture semantic meaning, on the Enron-Email dataset. The results are shown in Fig.11 (GPT-2) and Fig.12 (Llama-2). As illustrated in the figures, the performance of Random Label and gradient-ascent-based methods (Gradient Ascent and NegGrad+) is unstable under all utility metrics (Perplexity, BERTScore, and ROUGE). In contrast, Langevin Unlearning demonstrates relatively stable performance and achieves a favorable privacy-utility trade-off.

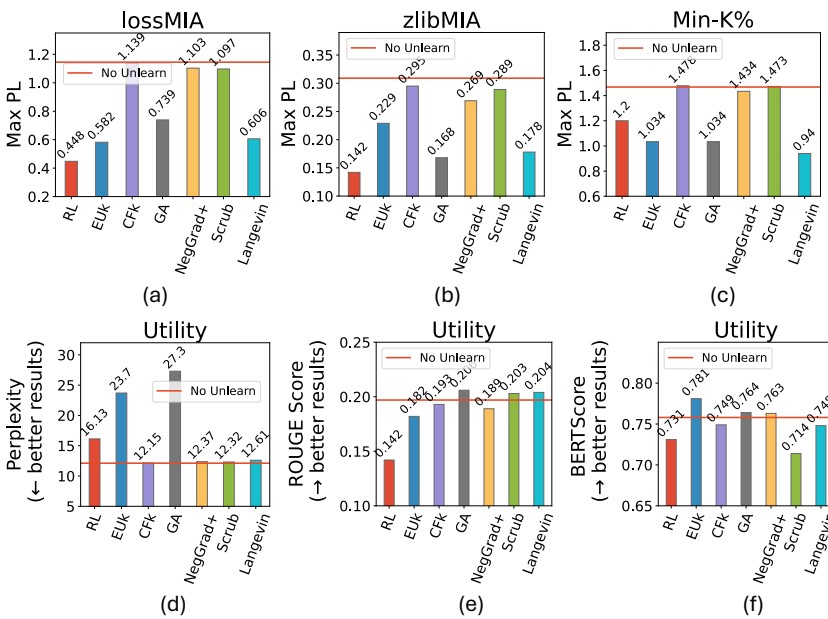

Figure 11: Benchmarking unlearning approaches via our minority-aware evaluation for GPT-2 on Enron-email dataset. (a)-(c): Maximum privacy leakage (PL) over three cases (Random, Canary, and Minority) for lossMIA, zlibMIA, and Min-K% attacks respectively. (d-f): Worst utility performance over the three cases of each method.

## C.4 FURTHER DETAILS AND RESULTS ON LANGEVIN UNLEARNING

In this section, we provide additional details and results on the Langevin Unlearning methods. As mentioned in Section 5, Langevin leverages noisy gradient descent and involves training the model on the dataset $D_{\text{train}}$ using DP-SGD. Furthermore, Langevin conducts machine unlearning by fine-tuning the model on the dataset $D_{\text{keep}}$ with DP-SGD as well.

It is important to note that for the Langevin Unlearning method, the training process incorporates noise. Consequently, our retrain baseline is adjusted to train the initial model on $D_{\text{keep}}$ using DP-SGD for 5 epochs. Furthermore, in Table 8 and 9, we report the effectiveness of Langevin Unlearning by evaluating it against three MIA methods (lossMIA, zlibMIA, and Min-K%) across different datasets

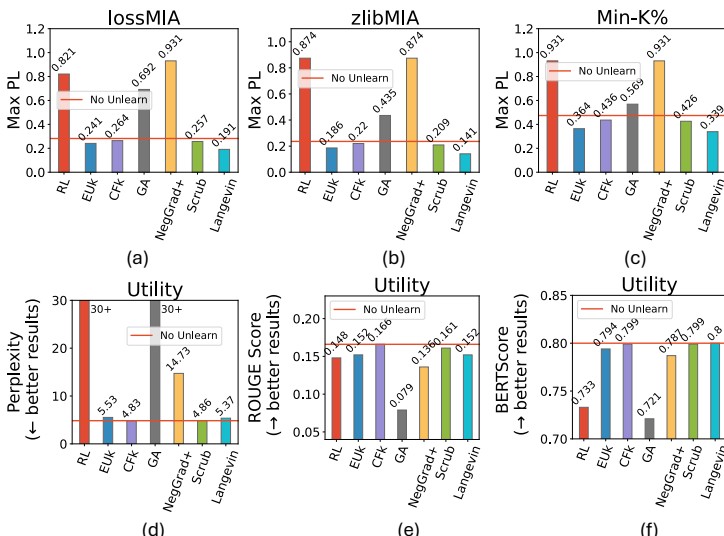

Figure 12: Benchmarking unlearning approaches via our minority-aware evaluation for Llama-2 on Enron-email dataset. (a)-(c): Maximum privacy leakage (PL) over three cases (`Random`, `Canary`, and `Minority`) for lossMIA, zlibMIA, and Min-K% attacks respectively. (d)-(f): Worst utility performance over the three cases of each method.

on GPT-2. These evaluations are conducted across three scenarios (`Random`, `Canary`, `Minority`), assessing the PL scores, the maximum PL scores and the worst-case perplexity. By comparing the results of the Noisy No Unlearn baseline (which fine-tunes the initial model with DP-SGD for 5 epochs) with those of the Langevin Unlearning method, we observe that minority scenarios (`Canary`, `Minority`) lead to significantly higher privacy leakage, and Langevin Unlearning achieves superior privacy-utility trade-offs. Additionally, in practical applications, the number of steps employing noisy gradient descent can be tailored based on the acceptable computational costs, thereby enabling potentially better privacy-utility trade-offs. This flexibility allows practitioners to balance the trade-off between enhanced privacy and computational efficiency according to specific application requirements.

Table 8: The privacy leakage (PL) for Langevin Unlearning against different attackers for GPT-2 on All datasets.

| Method | PL (lossMIA) | | | PL (zlibMIA) | | | PL (Min-K%) | | |
|---|---|---|---|---|---|---|---|---|---|
| | Random | Canary | Minority | Random | Canary | Minority | Random | Canary | Minority |
| | | | | Enron-phone | | | | | |
| Noisy No Unlearn | 0.097 | **0.152 (57%↑)** | **0.170 (75%↑)** | 0.024 | **0.039 (63%↑)** | **0.033 (38%↑)** | 0.164 | **0.259 (58%↑)** | **0.268 (63%↑)** |
| Langevin | 0.092 | **0.144 (57%↑)** | **0.157 (71%↑)** | 0.024 | **0.037 (54%↑)** | 0.027 (13%↑) | 0.159 | **0.259 (63%↑)** | **0.264 (66%↑)** |
| | | | | Enron-email | | | | | |
| Noisy No Unlearn | 0.156 | **0.342 (119%↑)** | **0.642 (312%↑)** | 0.102 | **0.193 (89%↑)** | **0.130 (27%↑)** | 0.344 | **0.691 (101%↑)** | **0.945 (175%↑)** |
| Langevin | 0.154 | **0.319 (107%↑)** | **0.606 (294%↑)** | 0.097 | **0.178 (84%↑)** | **0.124 (28%↑)** | 0.336 | **0.645 (92%↑)** | **0.939 (179%↑)** |
| | | | | ECHR-year | | | | | |
| Noisy No Unlearn | 0.101 | **0.152 (51%↑)** | **0.122 (21%↑)** | 0.049 | **0.067 (37%↑)** | **0.064 (31%↑)** | 0.122 | **0.180 (48%↑)** | 0.145 (19%↑) |
| Langevin | 0.103 | **0.140 (36%↑)** | **0.125 (21%↑)** | 0.049 | **0.061 (24%↑)** | **0.065 (33%↑)** | 0.117 | **0.168 (44%↑)** | **0.146 (25%↑)** |

## C.5 MORE RESULTS ON FORGET SET SIZE

In this section, we report additional results on the impact of forget set size for each unlearning method, using LossMIA and ZlibMIA attackers. As shown in Fig. 13, similar to the results in the main text, both RL and GA methods are sensitive to the forget set size, whereas methods like SCRUB and Langevin Unlearning demonstrate greater stability.

Table 9: Maximum PL Scores and Worst-case Perplexity for Noisy No Unlearn and Langevin across Datasets on GPT-2

| Dataset | Methods | lossMIA | zlibMIA | Min-K% | Perplexity |
|---------|---------|---------|---------|--------|------------|
| Enron phone | Noisy No Unlearn | 0.170 | 0.039 | 0.268 | 13.87 |
| | Langevin | 0.157 (7.65%↓) | 0.037 (5.13%↓) | 0.264 (1.49%↓) | 13.88 |
| Enron email | Noisy No Unlearn | 0.642 | 0.193 | 0.945 | 12.52 |
| | Langevin | 0.606 (5.61%↓) | 0.178 (7.77%↓) | 0.939 (0.63%↓) | 12.61 |
| ECHR year | Noisy No Unlearn | 0.152 | 0.067 | 0.180 | 12.75 |
| | Langevin | 0.140 (7.89%↓) | 0.061 (8.96%↓) | 0.168 (6.67%↓) | 12.78 |

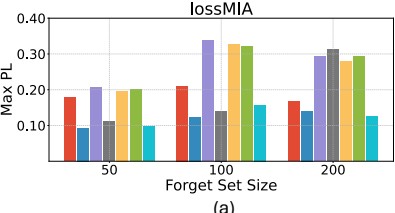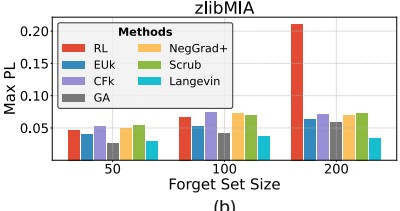

Figure 13: The effect of forget set size for each unlearning approach. (a)(b): Maximum PL over three cases (Random, Canary, Minority) with the attacker being lossMIA and zlibMIA respectively.

### C.6 MORE DETAILS ON THE PRIVACY-UTILITY TRADE-OFF CURVES FOR LANGEVIN UNLEARNING AND SCRUB METHODS

This section provides an comprehensive experiments of the privacy-utility trade-off curves for the Langevin Unlearning and SCRUB methods, as introduced in Sec. 6.3. We detailed the hyperparameters used in the ablation study as follows:

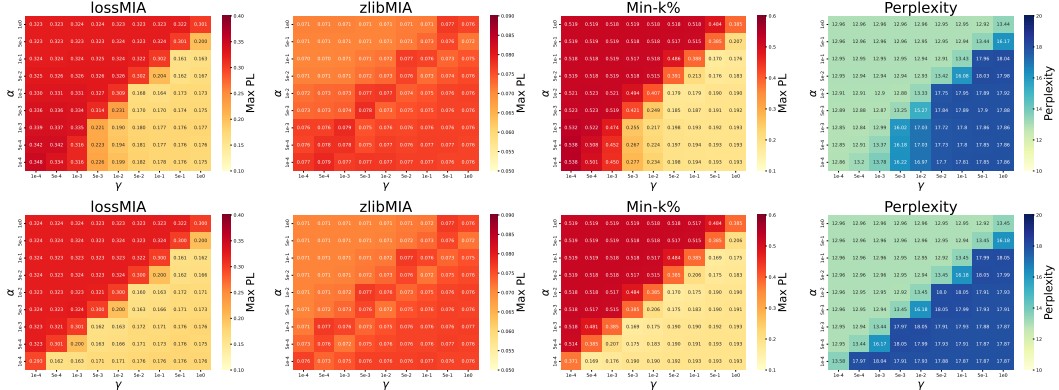

Figure 14: Privacy-utility transition curves for Enron-Phone dataset with hyperparameter $\beta = 1$ (Top) and hyperparameter $\beta = 1e - 3$ (Bottom).

**Langevin Unlearning.** For the Langevin Unlearning method, we fix the clipping norm to 1 and vary the noise scale added during training to control the privacy-utility trade-off. In experiments with the GPT-2 model, the noise scale $\sigma$ is adjusted across the values $\{1e - 4, 3e - 4, 5e - 4, 8e - 4, 1e - 3\}$. Table 18 and 19 present the AUC scores under various attackers (lossMIA, zlibMIA, Min-k%) and utility (perplexity) on the Enron-Phone and Enron-Email datasets, respectively.

**SCRUB.** The SCRUB training objective comprises the original loss $\ell$ on the keep set, along with two KL divergence regularizers on the keep and forget sets. These terms are balanced by three hyperparameters:

$$\hat{\mathbb{E}}_{x \sim D_{\text{keep}}}[\alpha \text{KL}(M_{\text{learn}}(x) \| M(x)) + \beta \ell(M; x)] - \hat{\mathbb{E}}_{x \sim D_{\text{forget}}}[\gamma \text{KL}(M_{\text{learn}}(x) \| M(x))]. \quad (2)$$

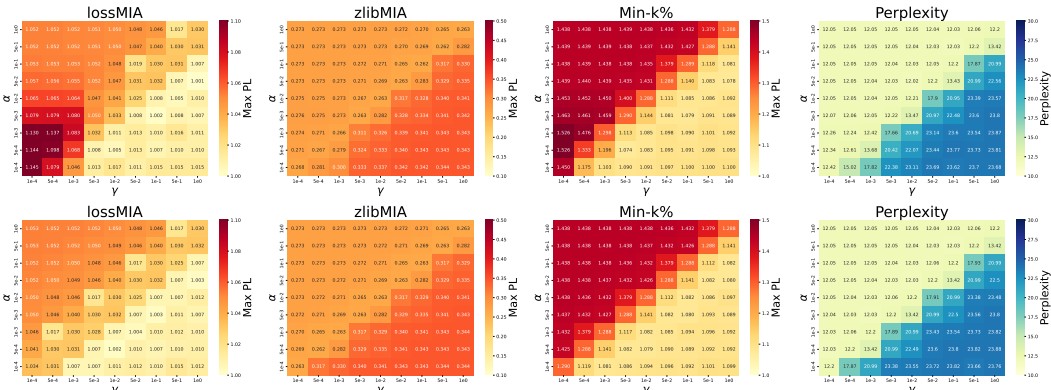

Figure 15: Privacy-utility transition curves for Enron-Email dataset with hyperparameter $\beta = 1$ (Top) and hyperparameter $\beta = 1e-3$ (Bottom).

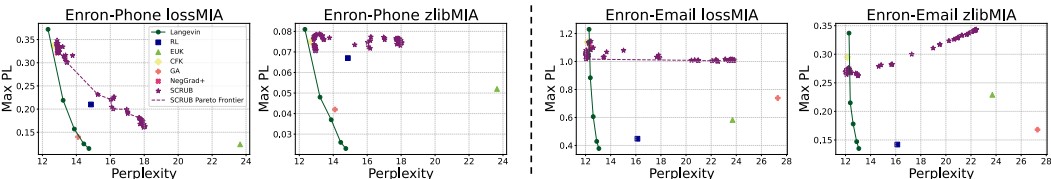

Figure 16: Privacy-utility trade-off curves under lossMIA and zlibMIA.

We conducted an extensive hyperparameter search, setting $\beta$ to 1 and $1e-3$ in separate configurations. For each fixed $\beta$, $\alpha$ and $\gamma$ are independently varied from $\{1e-4, 5e-4, 1e-3, 5e-3, 1e-2, 5e-2, 1e-1, 5e-1, 1\}$. Fig. 14 and 15 illustrates the resulting transition curves, showing the maximum privacy leakage (PL) for three scenarios (Random, Canary, Minority) across different attackers (lossMIA, zlibMIA, Min-k%) and utility (perplexity) metrics on both Enron-Phone and Enron-Email datasets. The transition curves highlight how SCRUB's performance depends on balancing the three objective terms. Notably, when the KL regularizer weight on the keep set is greater than or equal to that on the forget set, SCRUB achieves relatively high utility, albeit with increased privacy leakage. Besides, we observe in our experiments that the zlibMIA attacker fails to capture the inherent privacy-utility trade-off for SCRUB as demonstrated in the transition curves.

We further report the privacy-utility trade-off curves for both methods under attacker being lossMIA and zlibMIA. Similar to the results demonstrated in Sec. 6.3, Langevin Unlearning method achieves the best trade-off performance over SCRUB method.

## C.7 PRIVACY UNDERESTIMATION ACROSS COMPLEXITY UNITS

In Fig. 17, we report the degree of largest underestimation (`Canary` & `Minoirty` settings) in privacy leakage compared to `Random` settings across different complexity units under Min-k% attacker. As demonstrated in the figure, under different complexity units, the privacy leakage for each unlearning methods are severely underestimated. Detailed results under each settings are reported in Table 10 and 11.

## C.8 RESULTS ON AUC SCORES,
PERPLEXITY ACROSS DIFFERENT MODELS AND DATASETS.

We further report the AUC scores under different attackers (loss-MIA, zlibMIA, Min-K%) and utility (perplexity) over holdout test set $D_{\text{test}}$ for GPT-2 and Llama-2 in Table. 12-17.

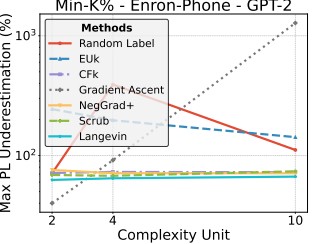

Figure 17: Degree of the largest underestimation (`Canary`, `Minority`) compared to Random settings across varying Complexity Units.

**Discussion on TPR@low FPR Metric:** Note that aside from
the AUC score, a commonly reported metric for privacy evaluation is **TPR@low FPR** (Carlini et al.,
2022), where the low FPR is often set to 0.01. However, in our scenario, the canary size is set to 100
(1% of the total training set size). At FPR = 0.01, the TPR would be calculated based on only a few
canary samples, making the overall score very coarse. To avoid the impact of this coarse granularity
on our experimental results, we primarily focus on the AUC score.

Table 10: PL Scores for MIA across Three Settings for GPT-2 on Enron (Phone) under Different
Complexity Units.

| | Enron-Phone GPT-2 Min-k% | | | | | | | | |
|---|---|---|---|---|---|---|---|---|---|
| Methods | Complexity Units 2 | | | Complexity Units 4 | | | Complexity Units 10 | | |
| | Random | Canary | Minority | Random | Canary | Minority | Random | Canary | Minority |
| Random Label | 0.166 | 0.284 | 0.136 | -0.059 | 0.037 | 0.290 | 0.168 | 0.320 | 0.355 |
| EUk | 0.068 | 0.169 | 0.234 | 0.068 | 0.174 | 0.202 | 0.092 | 0.216 | 0.223 |
| CFk | 0.302 | 0.445 | 0.518 | 0.300 | 0.438 | 0.519 | 0.298 | 0.435 | 0.514 |
| Gradient Ascent | -0.175 | -0.245 | -0.203 | -0.245 | -0.469 | -0.352 | -0.031 | -0.012 | -0.427 |
| NegGrad+ | 0.309 | 0.452 | 0.543 | 0.309 | 0.447 | 0.529 | 0.298 | 0.452 | 0.510 |
| Scrub | 0.311 | 0.443 | 0.525 | 0.313 | 0.447 | 0.525 | 0.306 | 0.445 | 0.532 |
| Langevin | 0.163 | 0.259 | 0.265 | 0.161 | 0.259 | 0.265 | 0.159 | 0.259 | 0.265 |

Table 11: Perplexity for MIA across Three Settings for GPT-2 on Enron (Phone) under Different
Complexity Units.

| | Enron-Phone GPT-2 Perplexity | | | | | | | | |
|---|---|---|---|---|---|---|---|---|---|
| Methods | Complexity Units 2 | | | Complexity Units 4 | | | Complexity Units 10 | | |
| | Random | Canary | Minority | Random | Canary | Minority | Random | Canary | Minority |
| Random Label | 29.85 | 28.93 | 30+ | 30+ | 30+ | 30+ | 30+ | 30+ | 30+ |
| EUk | 30+ | 30+ | 30+ | 30+ | 30+ | 30+ | 23.64 | 23.65 | 26.60 |
| CFk | 12.72 | 12.71 | 12.72 | 12.72 | 12.72 | 12.73 | 12.67 | 12.67 | 12.67 |
| Gradient Ascent | 16.63 | 15.76 | 29.28 | 30+ | 30+ | 30+ | 30+ | 30+ | 30+ |
| NegGrad+ | 12.86 | 12.87 | 12.83 | 12.84 | 12.86 | 12.88 | 12.83 | 12.80 | 12.88 |
| Scrub | 12.94 | 12.95 | 12.96 | 13.11 | 13.09 | 13.19 | 13.09 | 12.88 | 12.96 |
| Langevin | 13.84 | 13.85 | 13.88 | 13.85 | 13.86 | 13.88 | 13.84 | 13.88 | 13.88 |

Table 12: AUC Scores for MIA and Perplexity across Three Settings for GPT-2 on Enron (Phone Numbers)

| | AUC - LossMIA | | | AUC - ZlibMIA | | |
|---|---|---|---|---|---|---|
| | Random | Canary | Minority | Random | Canary | Minority |
| No Unlearn | 0.533 | 0.531 | 0.422 | 0.694 | 0.693 | 0.619 |
| Retrain | 0.448 | 0.414 | 0.315 | 0.660 | 0.644 | 0.582 |
| Noisy No Unlearn | 0.474 | 0.464 | 0.365 | 0.678 | 0.674 | 0.606 |
| Noisy Retrain | 0.432 | 0.403 | 0.312 | 0.662 | 0.649 | 0.587 |
| **Unlearning Methods** | | | | | | |
| Random Label | 0.501 | 0.493 | 0.381 | 0.689 | 0.687 | 0.617 |
| Langevin | 0.472 | 0.461 | 0.361 | 0.678 | 0.673 | 0.603 |
| EUk | 0.460 | 0.447 | 0.354 | 0.683 | 0.677 | 0.612 |
| CFk | 0.533 | 0.529 | 0.421 | 0.695 | 0.692 | 0.619 |
| Gradient Ascent | 0.488 | 0.472 | 0.355 | 0.676 | 0.671 | 0.597 |
| NegGrad+ | 0.530 | 0.526 | 0.418 | 0.694 | 0.691 | 0.616 |
| SCRUB | 0.523 | 0.518 | 0.416 | 0.692 | 0.689 | 0.618 |
| | AUC - Min-K% | | | Perplexity | | |
| | Random | Canary | Minority | Random | Canary | Minority |
| No Unlearn | 0.594 | 0.592 | 0.471 | 12.72 | 12.72 | 12.72 |
| Retrain | 0.457 | 0.409 | 0.309 | 12.74 | 12.74 | 12.74 |
| Noisy No Unlearn | 0.518 | 0.511 | 0.393 | 13.84 | 13.85 | 13.87 |
| Noisy Retrain | 0.445 | 0.406 | 0.310 | 13.84 | 13.84 | 13.83 |
| **Unlearning Methods** | | | | | | |
| Random Label | 0.575 | 0.573 | 0.447 | 14.49 | 14.41 | 14.86 |
| Langevin | 0.516 | 0.511 | 0.392 | 13.84 | 13.88 | 13.88 |
| EUk | 0.499 | 0.497 | 0.378 | 23.64 | 23.65 | 23.60 |
| CFk | 0.593 | 0.587 | 0.468 | 12.67 | 12.67 | 12.67 |
| Gradient Ascent | 0.526 | 0.508 | 0.362 | 13.22 | 13.20 | 14.10 |
| NegGrad+ | 0.591 | 0.587 | 0.467 | 12.86 | 12.86 | 12.88 |
| SCRUB | 0.592 | 0.593 | 0.472 | 13.00 | 12.98 | 12.96 |

Table 13: AUC Scores for MIA and Perplexity across Three Settings for GPT-2 on Enron (Email)

| | AUC - LossMIA | | | AUC - ZlibMIA | | |
|---|---|---|---|---|---|---|
| | Random | Canary | Minority | Random | Canary | Minority |
| **No Unlearn** | 0.555 | 0.551 | 0.354 | 0.462 | 0.462 | 0.563 |
| **Retrain** | 0.426 | 0.359 | 0.165 | 0.385 | 0.353 | 0.446 |
| **Noisy No Unlearn** | 0.488 | 0.483 | 0.271 | 0.422 | 0.421 | 0.512 |
| **Noisy Retrain** | 0.422 | 0.360 | 0.165 | 0.383 | 0.353 | 0.453 |
| | **Unlearning Methods** | | | | | |
| **Random Label** | 0.440 | 0.414 | 0.239 | 0.409 | 0.403 | 0.500 |
| **Langevin** | 0.487 | 0.475 | 0.265 | 0.420 | 0.416 | 0.509 |
| **EUk** | 0.525 | 0.517 | 0.261 | 0.437 | 0.434 | 0.514 |
| **CFk** | 0.552 | 0.544 | 0.353 | 0.461 | 0.457 | 0.562 |
| **Gradient Ascent** | 0.307 | 0.297 | 0.287 | 0.339 | 0.340 | 0.521 |
| **NegGrad+** | 0.539 | 0.528 | 0.347 | 0.454 | 0.448 | 0.558 |
| **SCRUB** | 0.548 | 0.538 | 0.346 | 0.458 | 0.455 | 0.559 |
| | AUC - Min-K% | | | Perplexity | | |
| | Random | Canary | Minority | Random | Canary | Minority |
| **No Unlearn** | 0.607 | 0.611 | 0.506 | 12.11 | 12.11 | 12.12 |
| **Retrain** | 0.397 | 0.316 | 0.205 | 12.19 | 12.19 | 12.19 |
| **Noisy No Unlearn** | 0.516 | 0.519 | 0.387 | 12.51 | 12.52 | 12.50 |
| **Noisy Retrain** | 0.384 | 0.307 | 0.199 | 12.48 | 12.48 | 12.48 |
| | **Unlearning Methods** | | | | | |
| **Random Label** | 0.568 | 0.560 | 0.451 | 15.87 | 16.13 | 13.54 |
| **Langevin** | 0.513 | 0.505 | 0.386 | 12.58 | 12.58 | 12.61 |
| **EUk** | 0.596 | 0.596 | 0.417 | 23.58 | 23.70 | 23.58 |
| **CFk** | 0.606 | 0.602 | 0.508 | 12.14 | 12.15 | 12.15 |
| **Gradient Ascent** | 0.242 | 0.220 | 0.417 | 27.30 | 21.86 | 14.53 |
| **NegGrad+** | 0.594 | 0.589 | 0.499 | 12.34 | 12.34 | 12.37 |
| **SCRUB** | 0.603 | 0.601 | 0.507 | 12.15 | 12.15 | 12.32 |

Table 14: AUC Scores for MIA and Perplexity across Three Settings for GPT-2 on ECHR (Year)

| | AUC - LossMIA | | | AUC - ZlibMIA | | |
|---|---|---|---|---|---|---|
| | Random | Canary | Minority | Random | Canary | Minority |
| **No Unlearn** | 0.661 | 0.650 | 0.658 | 0.532 | 0.524 | 0.560 |
| **Retrain** | 0.552 | 0.521 | 0.521 | 0.490 | 0.475 | 0.499 |
| **Noisy No Unlearn** | 0.600 | 0.592 | 0.581 | 0.514 | 0.509 | 0.535 |
| **Noisy Retrain** | 0.545 | 0.514 | 0.518 | 0.490 | 0.477 | 0.503 |
| **Unlearning Methods** | | | | | | |
| **Random Label** | 0.641 | 0.632 | 0.643 | 0.523 | 0.517 | 0.542 |
| **Langevin** | 0.601 | 0.586 | 0.583 | 0.514 | 0.506 | 0.536 |
| **EUk** | 0.621 | 0.613 | 0.593 | 0.523 | 0.517 | 0.534 |
| **CFk** | 0.656 | 0.643 | 0.656 | 0.531 | 0.520 | 0.559 |
| **Gradient Ascent** | 0.589 | 0.535 | 0.576 | 0.502 | 0.484 | 0.518 |
| **NegGrad+** | 0.653 | 0.636 | 0.650 | 0.525 | 0.517 | 0.555 |
| **SCRUB** | 0.651 | 0.637 | 0.653 | 0.529 | 0.522 | 0.557 |
| | AUC - Min-K% | | | Perplexity | | |
| | Random | Canary | Minority | Random | Canary | Minority |
| **No Unlearn** | 0.671 | 0.661 | 0.673 | 11.81 | 11.81 | 11.81 |
| **Retrain** | 0.553 | 0.518 | 0.518 | 11.82 | 11.82 | 11.82 |
| **Noisy No Unlearn** | 0.615 | 0.609 | 0.586 | 12.74 | 12.74 | 12.75 |
| **Noisy Retrain** | 0.548 | 0.516 | 0.512 | 12.73 | 12.73 | 12.73 |
| **Unlearning Methods** | | | | | | |
| **Random Label** | 0.658 | 0.652 | 0.651 | 12.70 | 12.70 | 12.67 |
| **Langevin** | 0.612 | 0.603 | 0.587 | 12.78 | 12.78 | 12.78 |
| **EUk** | 0.616 | 0.615 | 0.588 | 22.04 | 22.06 | 22.00 |
| **CFk** | 0.669 | 0.655 | 0.671 | 11.78 | 11.78 | 11.79 |
| **Gradient Ascent** | 0.603 | 0.508 | 0.592 | 12.11 | 12.20 | 12.11 |
| **NegGrad+** | 0.659 | 0.641 | 0.660 | 12.01 | 12.04 | 12.03 |
| **SCRUB** | 0.662 | 0.649 | 0.668 | 12.02 | 12.03 | 12.01 |

Table 15: AUC Scores for MIA and Perplexity across Three Settings for Llama-2 on Enron (Phone Number)

| | AUC - LossMIA | | | AUC - ZlibMIA | | |
|---|---|---|---|---|---|---|
| | **Random** | **Canary** | **Minority** | **Random** | **Canary** | **Minority** |
| **No Unlearn** | 0.614 | 0.606 | 0.551 | 0.578 | 0.571 | 0.575 |
| **Retrain** | 0.579 | 0.488 | 0.470 | 0.559 | 0.520 | 0.539 |
| **Noisy No Unlearn** | 0.605 | 0.598 | 0.539 | 0.578 | 0.572 | 0.576 |
| **Noisy Retrain** | 0.584 | 0.500 | 0.482 | 0.568 | 0.533 | 0.554 |
| **Unlearning Methods** | | | | | | |
| **Random Label** | 0.439 | 0.447 | 0.444 | 0.556 | 0.554 | 0.594 |
| **Langevin** | 0.603 | 0.590 | 0.532 | 0.577 | 0.569 | 0.574 |
| **EUk** | 0.612 | 0.608 | 0.557 | 0.581 | 0.575 | 0.583 |
| **CFk** | 0.612 | 0.603 | 0.549 | 0.577 | 0.569 | 0.573 |
| **Gradient Ascent** | 0.253 | 0.278 | 0.252 | 0.551 | 0.540 | 0.584 |
| **NegGrad+** | 0.536 | 0.398 | 0.451 | 0.547 | 0.495 | 0.538 |
| **SCRUB** | 0.613 | 0.567 | 0.550 | 0.578 | 0.553 | 0.574 |
| | AUC - Min-K% | | | Perplexity | | |
| | **Random** | **Canary** | **Minority** | **Random** | **Canary** | **Minority** |
| **No Unlearn** | 0.611 | 0.610 | 0.579 | 9.45 | 9.47 | 9.48 |
| **Retrain** | 0.568 | 0.547 | 0.491 | 9.48 | 9.48 | 9.48 |
| **Noisy No Unlearn** | 0.600 | 0.602 | 0.570 | 10.21 | 10.21 | 10.21 |
| **Noisy Retrain** | 0.579 | 0.565 | 0.516 | 10.19 | 10.19 | 10.19 |
| **Unlearning Methods** | | | | | | |
| **Random Label** | 0.498 | 0.507 | 0.497 | 2124 | 2559 | 2445 |
| **Langevin** | 0.598 | 0.596 | 0.563 | 10.20 | 10.21 | 10.20 |
| **EUk** | 0.604 | 0.619 | 0.584 | 11.00 | 11.24 | 10.82 |
| **CFk** | 0.609 | 0.606 | 0.575 | 9.43 | 9.45 | 9.46 |
| **Gradient Ascent** | 0.213 | 0.296 | 0.237 | 4e9 | 8e9 | 2e9 |
| **NegGrad+** | 0.529 | 0.399 | 0.463 | 9.82 | 9.90 | 9.85 |
| **SCRUB** | 0.610 | 0.530 | 0.578 | 9.45 | 9.48 | 9.48 |

Table 16: AUC Scores for MIA and Perplexity across Three Settings for Llama-2 on Enron (Email)

| | AUC - LossMIA | | | AUC - ZlibMIA | | |
|---|---|---|---|---|---|---|
| | **Random** | **Canary** | **Minority** | **Random** | **Canary** | **Minority** |
| **No Unlearn** | 0.628 | 0.613 | 0.418 | 0.548 | 0.539 | 0.451 |
| **Retrain** | 0.598 | 0.478 | 0.356 | 0.524 | 0.436 | 0.412 |
| **Noisy No Unlearn** | 0.594 | 0.572 | 0.393 | 0.515 | 0.502 | 0.443 |
| **Retrain** | 0.579 | 0.470 | 0.375 | 0.503 | 0.433 | 0.434 |
| | **Unlearning Methods** | | | | | |
| **Random Label** | 0.234 | 0.207 | 0.394 | 0.333 | 0.333 | 0.458 |
| **Langevin** | 0.592 | 0.560 | 0.393 | 0.513 | 0.494 | 0.443 |
| **EUk** | 0.620 | 0.593 | 0.416 | 0.532 | 0.517 | 0.454 |
| **CFk** | 0.627 | 0.604 | 0.416 | 0.548 | 0.532 | 0.449 |
| **Gradient Ascent** | 0.292 | 0.147 | 0.377 | 0.296 | 0.335 | 0.488 |
| **NegGrad+** | 0.041 | 0.034 | 0.064 | 0.088 | 0.055 | 0.215 |
| **SCRUB** | 0.622 | 0.601 | 0.418 | 0.542 | 0.527 | 0.451 |
| | AUC - Min-K% | | | Perplexity | | |
| | **Random** | **Canary** | **Minority** | **Random** | **Canary** | **Minority** |
| **No Unlearn** | 0.632 | 0.619 | 0.421 | 4.83 | 4.84 | 4.84 |
| **Retrain** | 0.594 | 0.420 | 0.344 | 4.84 | 4.84 | 4.84 |
| **Noisy No Unlearn** | 0.592 | 0.569 | 0.397 | 5.38 | 5.38 | 5.38 |
| **Noisy Retrain** | 0.570 | 0.410 | 0.368 | 5.39 | 5.39 | 5.39 |
| | **Unlearning Methods** | | | | | |
| **Random Label** | 0.230 | 0.181 | 0.330 | 730 | 542 | 255 |
| **Langevin** | 0.590 | 0.549 | 0.397 | 5.36 | 5.36 | 5.37 |
| **EUk** | 0.618 | 0.573 | 0.415 | 5.42 | 5.53 | 5.37 |
| **CFk** | 0.631 | 0.603 | 0.419 | 4.83 | 4.83 | 4.83 |
| **Gradient Ascent** | 0.256 | 0.219 | 0.445 | 4e8 | 6e12 | 6e12 |
| **NegGrad+** | 0.041 | 0.029 | 0.052 | 12.15 | 14.73 | 6.20 |
| **SCRUB** | 0.627 | 0.599 | 0.421 | 4.86 | 4.86 | 4.86 |

Table 17: AUC Scores for MIA and Perplexity across Three Settings for Llama-2 on ECHR (Year)

| | AUC - LossMIA | | | AUC - ZlibMIA | | |
|---|---|---|---|---|---|---|
| | Random | Canary | Minority | Random | Canary | Minority |
| **No Unlearn** | 0.570 | 0.547 | 0.726 | 0.513 | 0.499 | 0.513 |
| **Retrain** | 0.540 | 0.500 | 0.675 | 0.498 | 0.476 | 0.468 |
| **Noisy No Unlearn** | 0.556 | 0.532 | 0.721 | 0.504 | 0.491 | 0.510 |
| **Noisy Retrain** | 0.541 | 0.502 | 0.685 | 0.498 | 0.476 | 0.476 |
| **Unlearning Methods** | | | | | | |
| **Random Label** | 0.503 | 0.522 | 0.316 | 0.486 | 0.490 | 0.378 |
| **Langevin** | 0.555 | 0.528 | 0.715 | 0.503 | 0.488 | 0.498 |
| **EUk** | 0.572 | 0.542 | 0.728 | 0.513 | 0.497 | 0.505 |
| **CFk** | 0.570 | 0.544 | 0.724 | 0.512 | 0.497 | 0.509 |
| **Gradient Ascent** | 0.515 | 0.312 | 0.254 | 0.490 | 0.419 | 0.343 |
| **NegGrad+** | 0.553 | 0.364 | 0.254 | 0.504 | 0.429 | 0.358 |
| **SCRUB** | 0.570 | 0.547 | 0.724 | 0.513 | 0.499 | 0.510 |
| | AUC - Min-K% | | | Perplexity | | |
| | Random | Canary | Minority | Random | Canary | Minority |
| **No Unlearn** | 0.573 | 0.559 | 0.686 | 4.89 | 4.89 | 4.89 |
| **Retrain** | 0.537 | 0.502 | 0.603 | 4.89 | 4.89 | 4.89 |
| **Noisy No Unlearn** | 0.553 | 0.541 | 0.675 | 5.03 | 5.03 | 5.02 |
| **Noisy Retrain** | 0.537 | 0.504 | 0.616 | 5.02 | 5.02 | 5.02 |
| **Unlearning Methods** | | | | | | |
| **Random Label** | 0.519 | 0.537 | 0.327 | 90 | 129 | 127.43 |
| **Langevin** | 0.552 | 0.535 | 0.664 | 5.03 | 5.03 | 5.03 |
| **EUk** | 0.572 | 0.557 | 0.695 | 5.27 | 5.21 | 5.32 |
| **CFk** | 0.571 | 0.555 | 0.682 | 4.87 | 4.88 | 4.87 |
| **Gradient Ascent** | 0.503 | 0.299 | 0.257 | 7.94 | 10.68 | 29.48 |
| **NegGrad+** | 0.551 | 0.299 | 0.203 | 4.96 | 5.10 | 5.41 |
| **SCRUB** | 0.573 | 0.558 | 0.687 | 4.88 | 4.88 | 4.87 |

Table 18: AUC Scores for MIA and Perplexity across Three Settings for GPT-2 on Enron (Phone Number) for Noisy Learning

| | AUC - LossMIA | | | AUC - ZlibMIA | | |
|---|---|---|---|---|---|---|
| | **Random** | **Canary** | **Minority** | **Random** | **Canary** | **Minority** |
| **Noisy No Unlearn 1e-4** | 0.541 | 0.536 | 0.427 | 0.699 | 0.697 | 0.621 |
| **Noisy Retrain 1e-4** | 0.446 | 0.412 | 0.309 | 0.658 | 0.643 | 0.577 |
| **Langevin 1e-4** | 0.540 | 0.535 | 0.424 | 0.698 | 0.695 | 0.619 |
| **Noisy No Unlearn 3e-4** | 0.492 | 0.484 | 0.381 | 0.682 | 0.678 | 0.607 |
| **Noisy Retrain 3e-4** | 0.436 | 0.405 | 0.311 | 0.660 | 0.647 | 0.583 |
| **Langevin 3e-4** | 0.492 | 0.483 | 0.379 | 0.682 | 0.678 | 0.606 |
| **Noisy No Unlearn 5e-4** | 0.474 | 0.464 | 0.365 | 0.678 | 0.674 | 0.606 |
| **Noisy Retrain 5e-4** | 0.432 | 0.403 | 0.312 | 0.662 | 0.649 | 0.587 |
| **Langevin 5e-4** | 0.472 | 0.461 | 0.361 | 0.678 | 0.673 | 0.603 |
| **Noisy No Unlearn 8e-4** | 0.463 | 0.450 | 0.352 | 0.677 | 0.672 | 0.606 |
| **Noisy Retrain 8e-4** | 0.428 | 0.401 | 0.311 | 0.664 | 0.653 | 0.590 |
| **Langevin 8e-4** | 0.460 | 0.448 | 0.350 | 0.676 | 0.670 | 0.604 |
| **Noisy No Unlearn 1e-3** | 0.459 | 0.446 | 0.348 | 0.677 | 0.671 | 0.606 |
| **Noisy Retrain 1e-3** | 0.427 | 0.400 | 0.311 | 0.665 | 0.655 | 0.592 |
| **Langevin 1e-3** | 0.456 | 0.443 | 0.347 | 0.675 | 0.670 | 0.604 |
| | AUC - Min-K% | | | Perplexity | | |
| | **Random** | **Canary** | **Minority** | **Random** | **Canary** | **Minority** |
| **Noisy No Unlearn 1e-4** | 0.599 | 0.599 | 0.465 | 12.26 | 12.24 | 12.27 |
| **Noisy Retrain 1e-4** | 0.454 | 0.410 | 0.303 | 12.25 | 12.25 | 12.25 |
| **Langevin 1e-4** | 0.598 | 0.596 | 0.460 | 12.32 | 12.32 | 12.33 |
| **Noisy No Unlearn 3e-4** | 0.538 | 0.535 | 0.409 | 13.20 | 13.20 | 13.21 |
| **Noisy Retrain 3e-4** | 0.447 | 0.405 | 0.308 | 13.19 | 13.19 | 13.19 |
| **Langevin 3e-4** | 0.538 | 0.535 | 0.408 | 13.24 | 13.24 | 13.22 |
| **Noisy No Unlearn 5e-4** | 0.518 | 0.511 | 0.393 | 13.84 | 13.85 | 13.87 |
| **Noisy Retrain 5e-4** | 0.445 | 0.406 | 0.310 | 13.84 | 13.84 | 13.83 |
| **Langevin 5e-4** | 0.516 | 0.511 | 0.392 | 13.84 | 13.88 | 13.88 |
| **Noisy No Unlearn 8e-4** | 0.505 | 0.498 | 0.378 | 14.40 | 14.39 | 14.39 |
| **Noisy Retrain 8e-4** | 0.442 | 0.405 | 0.313 | 14.42 | 14.42 | 14.42 |
| **Langevin 8e-4** | 0.502 | 0.497 | 0.376 | 14.44 | 14.44 | 14.43 |
| **Noisy No Unlearn 1e-3** | 0.500 | 0.492 | 0.374 | 14.71 | 14.70 | 14.70 |
| **Noisy Retrain 1e-3** | 0.440 | 0.408 | 0.313 | 14.73 | 14.73 | 14.73 |
| **Langevin 1e-3** | 0.497 | 0.487 | 0.371 | 14.74 | 14.74 | 14.73 |

Table 19: AUC Scores for MIA and Perplexity across Three Settings for GPT-2 on Enron (Phone Email) for Noisy Learning

| | AUC - LossMIA | | | AUC - ZlibMIA | | |
|---|---|---|---|---|---|---|
| | **Random** | **Canary** | **Minority** | **Random** | **Canary** | **Minority** |
| **Noisy No Unlearn 1e-4** | 0.584 | 0.587 | 0.369 | 0.480 | 0.483 | 0.578 |
| **Noisy Retrain 1e-4** | 0.431 | 0.362 | 0.165 | 0.387 | 0.356 | 0.447 |
| **Langevin 1e-4** | 0.584 | 0.578 | 0.368 | 0.479 | 0.476 | 0.579 |
| **Noisy No Unlearn 3e-4** | 0.512 | 0.511 | 0.311 | 0.436 | 0.436 | 0.535 |
| **Noisy Retrain 3e-4** | 0.424 | 0.360 | 0.164 | 0.384 | 0.353 | 0.449 |
| **Langevin 3e-4** | 0.510 | 0.499 | 0.309 | 0.432 | 0.429 | 0.534 |
| **Noisy No Unlearn 5e-4** | 0.488 | 0.483 | 0.271 | 0.422 | 0.421 | 0.512 |
| **Noisy Retrain 5e-4** | 0.422 | 0.360 | 0.165 | 0.383 | 0.353 | 0.453 |
| **Langevin 5e-4** | 0.487 | 0.475 | 0.265 | 0.420 | 0.416 | 0.509 |
| **Noisy No Unlearn 8e-4** | 0.470 | 0.464 | 0.248 | 0.414 | 0.412 | 0.503 |
| **Noisy Retrain 8e-4** | 0.417 | 0.357 | 0.170 | 0.383 | 0.354 | 0.459 |
| **Langevin 8e-4** | 0.471 | 0.457 | 0.243 | 0.411 | 0.406 | 0.500 |
| **Noisy No Unlearn 1e-3** | 0.463 | 0.456 | 0.243 | 0.410 | 0.408 | 0.500 |
| **Noisy Retrain 1e-3** | 0.414 | 0.354 | 0.172 | 0.381 | 0.355 | 0.462 |
| **Langevin 1e-3** | 0.462 | 0.450 | 0.237 | 0.408 | 0.403 | 0.499 |
| | AUC - Min-K% | | | Perplexity | | |
| | **Random** | **Canary** | **Minority** | **Random** | **Canary** | **Minority** |
| **Noisy No Unlearn 1e-4** | 0.651 | 0.661 | 0.528 | 12.18 | 12.25 | 12.16 |
| **Noisy Retrain 1e-4** | 0.407 | 0.318 | 0.210 | 12.25 | 12.25 | 12.25 |
| **Langevin 1e-4** | 0.655 | 0.655 | 0.532 | 12.27 | 12.28 | 12.27 |
| **Noisy No Unlearn 3e-4** | 0.553 | 0.562 | 0.420 | 12.31 | 12.34 | 12.28 |
| **Noisy Retrain 3e-4** | 0.390 | 0.311 | 0.199 | 12.23 | 12.23 | 12.23 |
| **Langevin 3e-4** | 0.552 | 0.547 | 0.421 | 12.38 | 12.39 | 12.38 |
| **Noisy No Unlearn 5e-4** | 0.516 | 0.519 | 0.387 | 12.51 | 12.52 | 12.50 |
| **Noisy Retrain 5e-4** | 0.384 | 0.307 | 0.199 | 12.48 | 12.48 | 12.48 |
| **Langevin 5e-4** | 0.513 | 0.505 | 0.386 | 12.58 | 12.58 | 12.61 |
| **Noisy No Unlearn 8e-4** | 0.478 | 0.483 | 0.355 | 12.83 | 12.80 | 12.82 |
| **Noisy Retrain 8e-4** | 0.369 | 0.298 | 0.200 | 12.84 | 12.84 | 12.84 |
| **Langevin 8e-4** | 0.477 | 0.469 | 0.349 | 12.87 | 12.86 | 12.89 |
| **Noisy No Unlearn 1e-3** | 0.465 | 0.467 | 0.341 | 13.01 | 12.98 | 13.01 |
| **Noisy Retrain 1e-3** | 0.365 | 0.296 | 0.201 | 13.02 | 13.02 | 13.02 |
| **Langevin 1e-3** | 0.462 | 0.453 | 0.335 | 13.04 | 13.03 | 13.06 |

