# OpenReview forum: "Underestimated Privacy Risks for Minority Populations in Large Language Model Unlearning"
_ICLR.cc/2025/Workshop/BuildingTrust — Submitted to BuildingTrust_

### Official Review · Reviewer_b6Cn · 2025-03-02
**An insightful approach to audit the privacy of unlearning algorithms, but lack empirical validation across multiple random seeds**

**Rating:** 6
**Confidence:** 3

**Review:**

Summary:

This paper highlights that auditing the privacy of unlearning algorithms should not be done on any random forgotten samples because it can lead to underestimation of privacy risks, especially on high-risk samples such as those belonging to minority groups. In their experiments on three datasets and two models, the privacy leakage is found to be significantly larger on canary data and minority data, which corroborates their claim. Based on this finding, the paper proposes a minority-aware unlearning evaluation protocol by measuring the worst-case privacy leakage among three forget sets (i.e., random, canary, and minority). Through the proposed evaluation, the paper reveals that only the Langevin Unlearning method can achieve a good privacy-utility trade-off, surpassing other unlearning baselines that do not incorporate noise.

Overall, the paper is clearly written with reasonable motivation drawn from the privacy auditing literature and supporting empirical results on three datasets (Enron-Phone, Enron-Email, ECHR-Year) and two LLMs (GPT-2 and Llama-2). Although the results are only reported for a single random seed, I believe this concern can be addressed in the camera-ready version. Therefore, I would recommend acceptance with rating 6.

Pros:

1.⁠ ⁠The bias in selecting the forget set can greatly influence the performance of unlearning algorithms. Therefore, the problem targeted in this paper is important to ensure that unlearning algorithms and unlearned models are correctly evaluated from the privacy perspective.

2.⁠ ⁠The authors did a good job when providing results on many state-of-the-art unlearning algorithms in a controlled setting (same computational budget) to ensure fair comparison. The results from their experiments also have interesting implications when revealing that SOTA algorithms (e.g., SCRUB) are privacy-risky and encouraging algorithms with better privacy guarantees, such as Langevin Unlearning.

Cons:

1.⁠ ⁠From the empirical part, the results are not reliable enough as they are reported for a single random seed 42. The authors should provide supplementary results on multiple random seeds and their corresponding variance.

2.⁠ ⁠Based on Table 2 & 4-6, GA and RL incur less privacy leakage for canaries and minority data. The authors could provide potential explanations and if possible, empirical results to confirm their explanations.

3.⁠ ⁠Based on the experiment description in Section 6, the authors used 10k samples for GPT-2 experiments and 50k samples for Llama-2 experiments. I’m wondering why the authors didn’t use the same dataset size for both experiments and didn’t use the entire dataset.

4.⁠ ⁠Although I agree that evaluating unlearning algorithms on random subsets can give a false sense of privacy, the idea of measuring worst-case privacy on minority groups is well-known in the privacy auditing literature. Therefore, the novelty of the findings in this paper is limited in my opinion.

Suggestions:

1.⁠ ⁠The authors should provide supplementary results on other random seeds.

2.⁠ ⁠The authors may include results on the full datasets or provide explanations for their choice of dataset resampling.

3.⁠ ⁠As the proposed evaluation protocol is evaluated on minority groups, the authors can also discuss the approach to choosing the minority groups in real-world datasets instead of relying on PII datasets.

---

### Official Review · Reviewer_iooh · 2025-03-02
**The paper highlights the importance of a minority-aware approach in LLM unlearning and proposes a novel evaluation protocol, its findings are undermined by methodological shortcomings, including concerns over dataset validity and the omission of stronger MIA techniques.**

**Rating:** 6
**Confidence:** 3

**Review:**

This work underlines the importance of a minority-aware approach in evaluating LLM unlearning and shows the disproportionate impact of privacy leakage on minority groups. The proposed evaluation protocol addresses this gap, enabling a more comprehensive assessment of unlearning methods. The findings underscore the significance of incorporating noise in unlearning approaches.

However, the paper has several notable shortcomings:

- The use of the ECHR dataset raises questions about the validity of the  (MIA) results. If all partitions of the dataset were used, it could lead to significantly better MIA performance due to the temporal shift between train and test splits. Recent literature has highlighted this issue, showing that MIA can work much better under these conditions [1], and even blind models can distinguish members from non-members [2].

- The study overlooks stronger, reference-free MIA techniques that could have provided more robust results. Notably, it fails to consider advanced methods such as Min-K%++ [3] and CAMIA (Context-Aware Membership Inference Attack) [4].

-  The paper in some points does not cite the mentioned methods (e.g. lines 99-100).

[1] Zhang, J., Das, D., Kamath, G., & Tramèr, F. (2024). Membership inference attacks cannot prove that a model was trained on your data. arXiv preprint arXiv:2409.19798.
[2] Das, D., Zhang, J., & Tramèr, F. (2024). Blind baselines beat membership inference attacks for foundation models. arXiv preprint arXiv:2406.16201.
[3] Zhang, J., Sun, J., Yeats, E., Ouyang, Y., Kuo, M., Zhang, J., ... & Li, H. (2024). Min-k%++: Improved baseline for detecting pre-training data from large language models. arXiv preprint arXiv:2404.02936.
[4] Chang, H., Shamsabadi, A. S., Katevas, K., Haddadi, H., & Shokri, R. (2024). Context-aware membership inference attacks against pre-trained large language models. arXiv preprint arXiv:2409.13745.

---

### Decision · Program_Chairs · 2025-03-05

Reject